# Repurposing of KLF5 activates a cell cycle signature during the progression from a precursor state to oesophageal adenocarcinoma

Connor Rogerson[1], Samuel Ogden[1], Edward Britton[1], The OCCAMS Consortium, Yeng Ang[2,3]*, Andrew D Sharrocks[1]*

[1]School of Biological Sciences, Faculty of Biology, Medicine and Health, University of Manchester, Manchester, United Kingdom; [2]School of Medical Sciences, Faculty of Biology, Medicine and Health, University of Manchester, Manchester, United Kingdom; [3]GI Science Centre, Salford Royal NHS FT, University of Manchester, Salford, United Kingdom

*For correspondence:
Yeng.Ang@srft.nhs.uk (YA);
andrew.d.sharrocks@manchester.ac.uk (ADS)

Competing interests: The authors declare that no competing interests exist.

**Abstract** Oesophageal adenocarcinoma (OAC) is one of the most common causes of cancer deaths. Barrett's oesophagus (BO) is the only known precancerous precursor to OAC, but our understanding about the molecular events leading to OAC development is limited. Here, we have integrated gene expression and chromatin accessibility profiles of human biopsies and identified a strong cell cycle gene expression signature in OAC compared to BO. Through analysing associated chromatin accessibility changes, we have implicated the transcription factor KLF5 in the transition from BO to OAC. Importantly, we show that KLF5 expression is unchanged during this transition, but instead, KLF5 is redistributed across chromatin to directly regulate cell cycle genes specifically in OAC cells. This new KLF5 target gene programme has potential prognostic significance as high levels correlate with poorer patient survival. Thus, the repurposing of KLF5 for novel regulatory activity in OAC provides new insights into the mechanisms behind disease progression.

## Introduction

Oesophageal cancer is the eighth most common cancer worldwide, and its 5-year survival rate of 15% makes it the sixth most-common cause of cancer-related death (*Ferlay et al., 2015*; *Pennathur et al., 2013*). A subtype of oesophageal cancer, oesophageal adenocarcinoma (OAC), is the predominant subtype in many Western countries and its incidence is rising rapidly (*Coleman et al., 2018*). Patients with OAC often present at a late stage with advanced disease (*Smyth et al., 2017*). The lack of molecular knowledge of OAC, combined with lack of tailored therapies, contribute to the low survival of OAC patients.

The accepted model of OAC development is the progression from an intestinal metaplastic condition of the lower oesophagus, known as Barrett's oesophagus (BO), to OAC through increasing stages of dysplasia (*Burke and Tosh, 2012*; *Spechler and Souza, 2014*). Many mutations found in OAC are also present in BO, especially *TP53*, which suggests a stepwise transition to OAC (*Ross-Innes et al., 2015*; *Stachler et al., 2015*). Focal amplifications differ as they largely occur in OAC compared to BO (*Lin et al., 2012*; *Stachler et al., 2015*; *Yamamoto et al., 2016*). The amplified genes can be grouped into functional biological pathways with the RAS-ERK signalling pathway (e.g. ERBB2; EGFR; KRAS) and GATA transcription factors (GATA4; GATA6) being the most common (*Frankell et al., 2019*; *Lin et al., 2012*; *Cancer Genome Atlas Research Network et al., 2017*). The morphology of BO differs from the oesophageal epithelia by the presence of a columnar epithelium

**eLife digest** Acid fluids present in the gut can sometimes 'go up' and damage the oesophagus, the pipe that connects the mouth and the stomach. As a result, a small number of individuals can develop Barrett's oesophagus, a condition where cells in the lining of the lower oesophagus show abnormal shapes. In certain patients, these cells then become cancerous, but exactly how this happens is unknown. This lack of understanding contributes to late diagnoses, limited treatment and low survival rates.

Many cancers feature 'signature' mutations in a set of genes that controls how a cell can multiply. Yet, in the case of cancers of the lower oesophagus, known genetic changes have had a limited impact on our understanding of the emergence of the disease. Here, Rogerson et al. focused instead on non-genetic changes and studied transcription factors, the proteins that bind to regulatory regions of the DNA to switch genes on and off.

A close inspection of cancer cells in the lower oesophagus revealed that, in that state, a transcription factor called KLF5 controls the abnormal activation of genes involved in cell growth. This is linked to the transcription factor adopting a different pattern of binding onto regulatory regions in diseased cells. Crucially, when the cell growth genes regulated by KLF5 are activated, patients have lower survival rates. Further work is now required to examine whether this finding could help to identify patients who are most at risk from developing cancer. More broadly, the results from the work by Rogerson et al. demonstrate how transcription factors can be repurposed in a disease context.

and secretory goblet cells, rather than squamous epithelium (reviewed in *Spechler and Souza, 2014*). Genomic and transcription events have been observed to differ between BO and OAC. Mutations in *TP53* are more frequent in BO from patients that had progressed to OAC (*Stachler et al., 2018*) and *SMAD4* mutations appear to occur exclusively in OAC, although at a low frequency (*Weaver et al., 2014*). Increased TGFβ signalling through other SMAD family members, SMAD2/3, promotes growth in OAC cells (*Blum et al., 2019*). Additionally, increased expression and increased activity of AP-1 transcription factors occurs in the transition from BO to OAC (*Blum et al., 2019*; *Britton et al., 2017*; *Maag et al., 2017*). Despite these studies, the definitive molecular mechanisms of progression to OAC are poorly understood and biomarkers to identify patients at risk of progression are lacking.

Changes to the chromatin landscape have been implicated in many cancers and chromatin accessibility changes during tumourigenesis are a major factor in altering regulatory element activity (*Britton et al., 2017*; *Corces et al., 2018*; *Davie et al., 2015*; *Denny et al., 2016*; *Kelso et al., 2017*; *Rendeiro et al., 2016*; *Tome-Garcia et al., 2018*; *Zhou and Guo, 2018*). We recently used Assay for Transposase-Accessible Chromatin using sequencing (ATAC-seq) to ascertain the molecular basis of BO and identified a set of transcription factors that define the BO chromatin landscape and are retained in OAC (*Rogerson et al., 2019*). Here, we took a similar approach to discover important transcriptional regulators (*Figure 1A*) that are specifically operational in OAC and hence contribute to the molecular basis of disease progression from BO to OAC. We compared the open chromatin landscape in BO and OAC patient biopsies and uncovered KLF5 as an important transcriptional regulator that is repurposed to directly drive a cell cycle gene expression signature during the progression of BO to OAC.

## Results

### Enhanced cell cycle activity defines BO progression to OAC

To begin to understand the molecular events that distinguish OAC form the BO precursor state we first established the differential gene expression profiles between BO and OAC. We analysed public human BO and OAC RNA-seq data (*Maag et al., 2017*). These samples separate well after principal component analysis (PCA), therefore we retained all samples for further analysis (*Figure 1—figure supplement 1A*). Performing differential gene expression analysis, we identified 905 differentially expressed genes between BO and OAC (±1.5 x; *Q*-value <0.05; *Figure 1B*; *Supplementary file 1*).

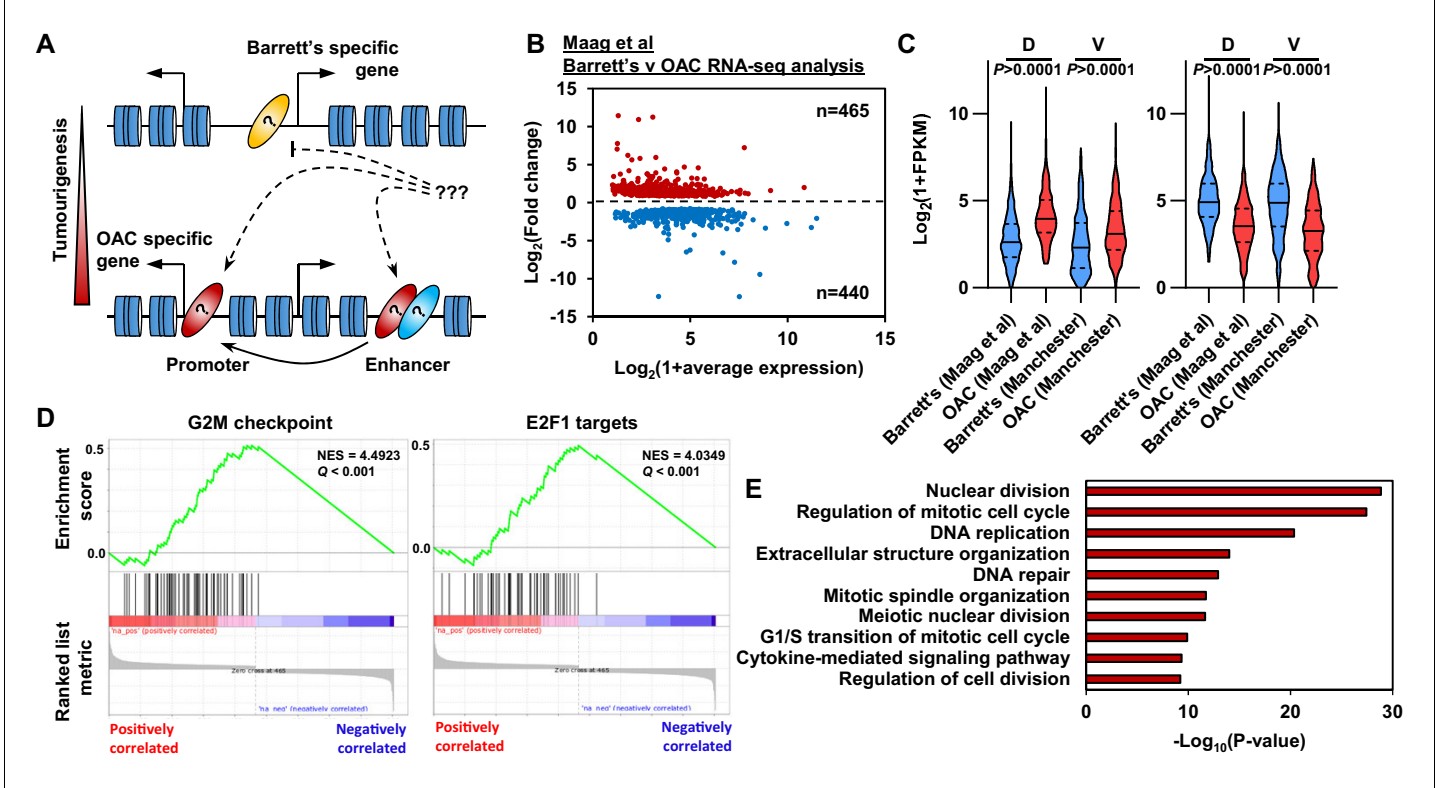

**Figure 1.** Oesophageal adenocarcinoma (OAC) tumourigenesis is associated with enhanced cell cycle gene activity. (**A**) Schematic of possible transcription factor network induction during OAC development. Coloured ovals represent different transcription factors. (**B**) Scatter plot of significant differentially (±1.5 x, *Q*-value <0.05) expressed genes between human Barrett's oesophagus n = 13 and human OAC n = 12 samples (*Maag et al., 2017*). (**C**) Violin plots of expression of differentially expressed genes between Barrett's oesophagus (n = 13) and oesophageal adenocarcinoma (n = 12) from discovery dataset (D; *Maag et al., 2017*) and validation dataset (V; BO = 3; OAC n = 4). Genesets are shown for upregulated (left) and downregulated (right) in OAC. (**D**) Gene set enrichment analysis of differentially expressed genes. The top two upregulated gene sets are shown with normalised enrichment score (NES) and *Q*-value. (**E**) Biological pathway GO term analysis of upregulated genes. The top 10 terms are shown. See also *Figure 1—figure supplement 1*.

The online version of this article includes the following figure supplement(s) for figure 1:

**Figure supplement 1.** Differential gene expression analysis between Barrett's and OAC patient samples.

Of these 905 genes, 465 are upregulated in OAC and 440 are downregulated in OAC compared to BO. To validate these findings, we analysed RNA-seq data from our own sample collection (3 BO and 3 OAC). Genes that were upregulated in OAC from the discovery dataset were significantly upregulated in the validation dataset and likewise for downregulated in OAC genes (*Figure 1C*). To gain insights into biological pathways behind these differentially expressed genes, we used two approaches. Firstly, Gene Set Enrichment Analysis (GSEA) uncovered two cell cycle associated terms, 'G2M checkpoint' and 'E2F1 targets', as the most significant upregulated gene sets in OAC (*Figure 1D*). Conversely, 'Fatty acid metabolism' and 'p53 pathway' are the most significant downregulated gene sets (*Figure 1—figure supplement 1B*). Secondly, biological pathway gene ontology analysis of upregulated genes revealed many cell cycle associated terms, such as 'Nuclear division', 'Regulation of mitotic cell cycle' and 'DNA replication' (*Figure 1E*). Example genes such as *CDC25B*, *CENPI* and *E2F1* all showed significant upregulation in OAC compared to BO in both datasets (*Figure 1—figure supplement 1D*). Downregulated genes uncovered metabolic associated terms, such as 'alcohol metabolic process', 'monocarboxylic acid metabolic process' and 'Lipid catabolic process' (*Figure 1—figure supplement 1C*). Representative example genes from these pathways such as *IDI1*, *ADH4* and *CIDEC* all show significant downregulation in both datasets (*Figure 1—figure supplement 1E*). These initial results indicate a strong upregulation of genes associated with cell

cycle processes during the progression from BO to OAC accompanied with the inactivation of genes controlled by the p53 pathway and genes associated with metabolism.

## Chromatin accessibility changes in the transition from BO to OAC

To identify putative transcriptional regulators that may drive the transition to OAC and impact on this enhanced cell cycle profile, we analysed the accessible chromatin landscape using ATAC-seq from patient biopsies. To supplement our previous ATAC-seq datasets from BO and OAC patients (*Britton et al., 2017*; *Rogerson et al., 2019*), we performed ATAC-seq on two additional OAC biopsies, which were quality-checked and reproducible (*Figure 2—figure supplement 1A and B*). We wanted to focus on the differentially expressed genes in OAC compared to BO, therefore we generated a set of accessible regions representing potential regulatory regions that are associated with this set of genes. We took all ATAC-seq peaks from all samples within +/- 250 kb of a TSS of a differentially expressed gene (*Figure 2A*). After merging so that only unique peaks remained, 35,220 regions were used for further analyses (*Supplementary file 2*). We first performed principal component analysis on normalised ATAC-seq signal of all BO and OAC samples to identify differences between samples (*Figure 2B*). This led to clustering of all BO samples and clustering of most OAC samples. OAC samples T_003 and T_005 did not cluster with the other OAC samples and were therefore removed from the subsequent differential accessibility analysis. We then carried out differential accessibility analysis between BO and OAC on this peak set (*Figure 2C*; *Supplementary file 2*). A total of 1495 regions were significantly differentially accessible (±2 x; *Q*-value <0.1), the majority of which increased in accessibility (1327/1495). An example gene locus which shows differential accessibility in OAC is centred on *KRT19* (*Figure 2D*). Within this locus, both gene promoters (14%; 1/7) and distal regulatory regions (86%; 6/7) gain accessibility in OAC. To assess whether the observed changes of accessibility near differentially expressed genes are common to other OACs, we compared our ATAC-seq data to independent, previously published ATAC-seq datasets from TCGA-ESCA oesophageal adenocarcinoma samples (*Figure 2D*, bottom; *Figure 2E*; *Corces et al., 2018*). TCGA-ESCA samples showed similar open chromatin peak profiles and clustered with our OAC samples with the exception of one sample, which clusters with our BO samples (*Figure 2E*). The chromatin accessibility profiles nearby genes differentially expressed in OAC are therefore reproducible across patients.

Next, we harnessed the differential accessibility data to uncover the identities of transcription factors bound in these regions. De novo motif discovery of regions that become more accessible in OAC contain significantly enriched motifs for AP-1, KLF, TBX, NFκB and p53 transcription factor families (*Figure 3A*; *Supplementary file 3*). AP-1 and KLF were clearly the most frequent motifs in the differential regions and showed the strongest match score for the consensus motif. Regions that showed decreased chromatin accessibility in OAC are enriched in EWSR1-FLI1, ASCL2, GLI2, E2F and ZBTB18 motifs, albeit with relatively low match scores (*Figure 2—figure supplement 1C*; *Supplementary file 3*). To further assess which transcription factors might be involved in gene expression control, we carried out footprinting analysis on differential accessible regions from our ATAC-seq datasets (*Figure 3B*; *Bentsen et al., 2020*). In differential accessible regions, motifs for KLF (e.g. KLF4, KLF5 and KLF1) and AP-1 (e.g. FOS, JUNB, JUND, JUN and FOSL1/2) transcription factors showed the highest footprinting score in OAC, whereas motifs for homeobox transcription factors (e.g. HNF1A, HOXA5 and NKX2-5), ARID3A and MEF transcription factors (e.g. MEF2A and MEF2C) showed more footprinting in BO. To provide more evidence for transcription factor occupancy, we then plotted ATAC-seq signal across their motifs. Both FOS (AP-1) and KLF4 (KLF) motifs show a clear increase in footprint depth in OAC, indicative of more transcription factor binding (*Figure 3C*). We have previously identified AP-1 as an important regulator in OAC (*Britton et al., 2017*), but the role of KLF transcription factors in OAC is poorly understood. We therefore focussed on the potential role of KLF transcription factors in the progression of BO to OAC.

## KLF5 controls expression of cell cycle genes in OAC

To identify a specific KLF transcription factor that may be bound to these accessible regions, we analysed the expression of individual KLF transcription factors in OAC samples (*Figure 3D*). KLF5 was clearly the highest expressed among the KLF family in OAC. KLF5 has been previously implicated in oesophageal squamous cell carcinoma as a tumour suppressor (*Tarapore et al., 2013*) and has been

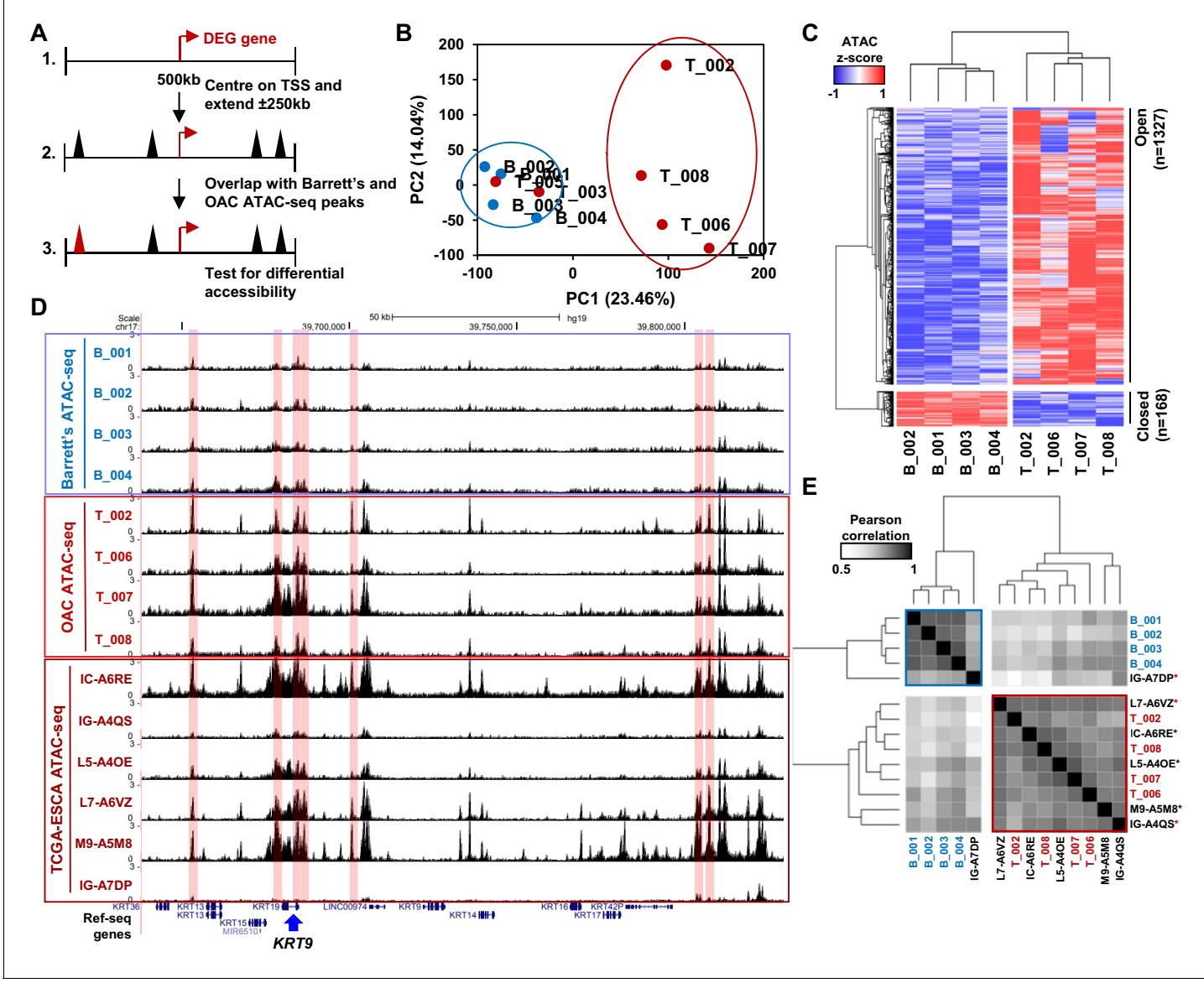

**Figure 2.** Altered chromatin accessibility landscape during OAC carcinogenesis. (A) Schematic of ATAC-seq analysis. All peaks within ±250 kb of the TSS of a differentially expressed genes were assessed for differential accessibility between Barrett's oesophagus and OAC. (B) Principal Component Analysis plot of $\log_2$(1+FPKM) ATAC-seq signal from all accessible regions within ±250 kb of a differentially expressed gene TSS from all human Barrett's oesophagus (B; n = 4) and oesophageal adenocarcinoma samples (T; n = 6). (C) Heatmap of z-score ATAC-seq signal from human Barrett's oesophagus (B; n = 4) and OAC (T; n = 4) samples at differentially accessible regions (±2 x; Q < 0.1). Hierarchical clustering of samples and regions performed using 1-Pearson correlation. (D) Example UCSC browser view of BO, OAC and TCGA ESCA ATAC-seq data surrounding the *KRT19* locus with differentially accessible regions highlighted in red. (E) Correlation plot of Pearson correlation of $\log_2$(1+FPKM) ATAC-seq signal at differentially accessible regions. Hierarchical clustering performed using 1-Pearson correlation and the two main clusters are highlighted blue (BO) and red (OAC). TCGA samples are indicated by asterisks. See also *Figure 2—figure supplement 1*.

The online version of this article includes the following figure supplement(s) for figure 2:

**Figure supplement 1.** ATAC-seq analysis of patient OAC samples.

identified as pro-tumorigenic in gastric cancer via amplifications (*Chia et al., 2015*). To determine the gene regulatory functions of KLF5, we carried out siRNA-mediated knockdowns of KLF5 in OE19 cells, a cell line we identified as having a similar chromatin landscape to OAC biopsies (*Rogerson et al., 2019*) and exhibits strong tumourigenic properties (*Hassan et al., 2017*). Knockdown of KLF5 was evident after 3 days siRNA transfection (*Figure 3—figure supplement 1A*) and

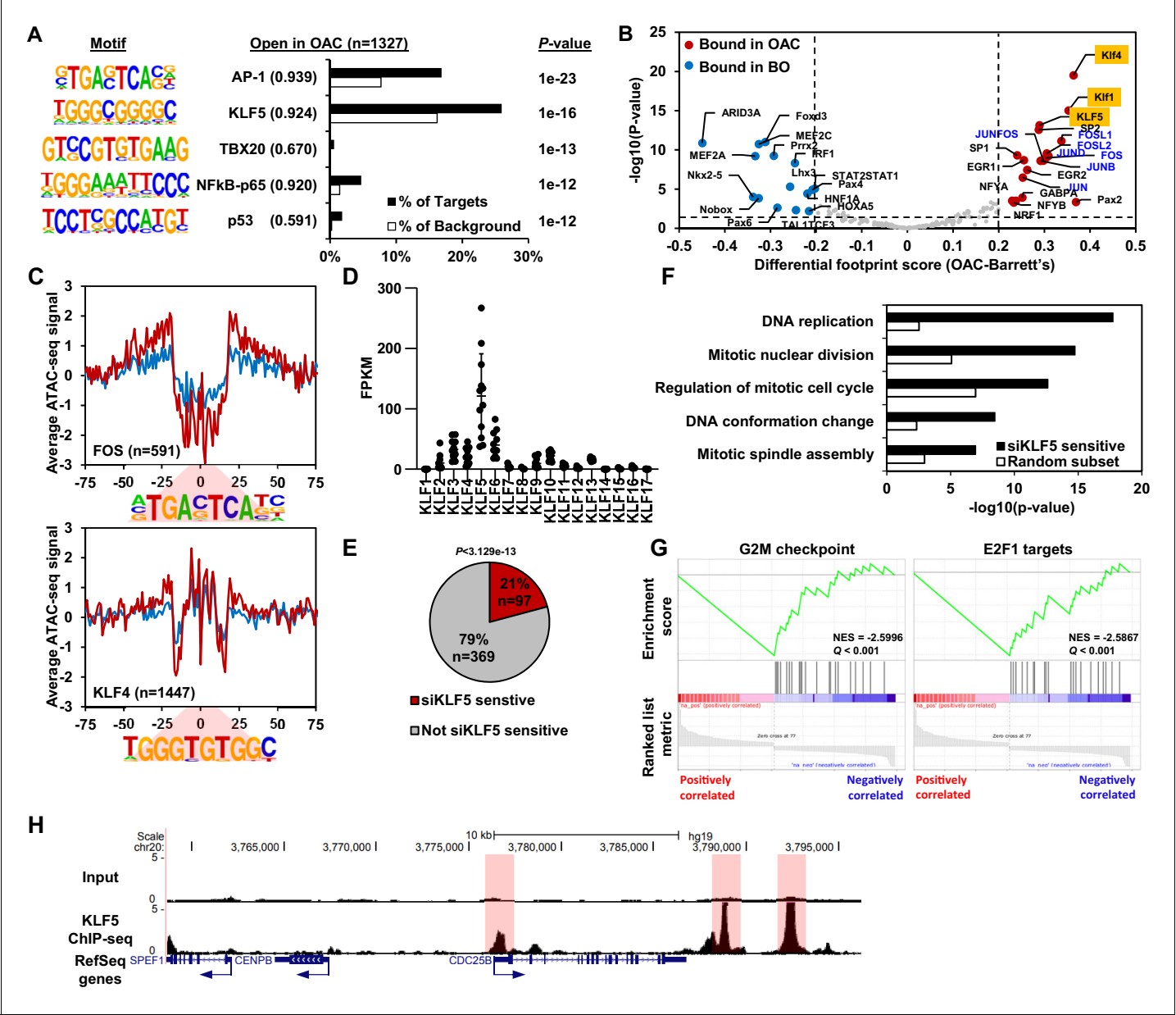

**Figure 3.** KLF5 control a cell cycle gene expression programme in OAC. (**A**) Bar chart of percentage targets and percentage background of de novo discovered motifs at increased accessible regions in OAC compared to Barrett's oesophagus. De novo motif, called transcription factor with motif match score (brackets) and *P*-value are shown. (**B**) Scatter plot of differential footprinting depth around human transcription factor motifs in differential accessible regions in BO and OAC tissue. Significant motifs with more footprint depth in BO are labelled blue and in OAC labelled red. KLF TF-binding motifs are highlighted in orange and AP1 motifs are in blue font. (**C**) BO (blue) and OAC (red) ATAC-seq signal at FOS (AP1) and KLF4 (KLF) motifs in differentially accessible regions. (**D**) Expression (FPKM) of KLF family transcription factors in OAC RNA samples (n = 12; *Maag et al., 2017*). (**E**) Pie chart of percentage of upregulated genes in patient OAC samples that are also downregulated with siKLF5 treatment. p-Value shown. (**F**) Biological pathway GO term analysis of OAC upregulated and siKLF5 downregulated genes and a random gene selection. (**G**) Gene set enrichment analysis of genes that are upregulated in OAC and downregulated by siKLF5 treatment. Top two downregulated gene sets are shown. Normalised enrichment scores (NES) and *Q*-values are shown. (**H**) Example UCSC Genome Browser view of KLF5 ChIP-seq binding at the *CDC25B* locus. KLF5 peaks highlighted in red. See also *Figure 3—figure supplements 1* and *2*.

The online version of this article includes the following figure supplement(s) for figure 3:

**Figure supplement 1.** Identification of the KLF5-regulated cistrome in OE19 cells.

**Figure supplement 2.** KLF5 ChIP-seq analysis in OE19 cells.

RNA-seq replicates were highly correlative (*Figure 3—figure supplement 1B*). Carrying out differential expression analysis identified 4934 genes (2637 upregulated and 2297 downregulated) with significant changes in gene expression (±1.3 x; Q-value <0.05; *Figure 3—figure supplement 1C*; *Supplementary file 4*). Biological pathway GO term analysis revealed several enriched terms including 'DNA replication' and 'Regulation of mitotic cell cycle' for downregulated genes, and terms involving 'Oxidative phosphorylation' and 'mitochondrial gene expression' for upregulated genes (*Figure 3—figure supplement 1D and E*). The terms associated with downregulated genes are reminiscent of the terms enriched in genes upregulated in OAC (see *Figure 1*). Moreover, GSEA also found similar gene sets: 'mitotic spindle'; 'G2M checkpoint' and 'E2F targets' for downregulated genes and 'oxidative phosphorylation'; 'xenobiotic metabolism' and 'fatty acid metabolism' for upregulated genes (*Figure 3—figure supplement 1F and G*). Since the genes regulated by KLF5 are involved in similar processes as the genes aberrantly expressed in OAC, we asked whether any of the same genes are in each dataset. 21% (97/465) of the genes upregulated in OAC significantly overlap with those downregulated with siKLF5 (*Figure 3E*) and many of these are associated with cell cycle related functions, including genes encoding core cell cycle proteins like CCNE1, E2F1 and various MCM proteins (*Figure 3—figure supplement 1H*). Further analysis of the biological pathways enriched within these 97 genes identified very similar GO terms to those enriched in genes upregulated in OAC compared to BO (*Figure 3F*). GSEA also identified the same gene set terms: 'G2M checkpoint' and 'E2F1 targets' (*Figure 3G*).

Next, we asked whether these genes are directly regulated by KLF5, and carried out replicate ChIP-seq for KLF5 in OE19 cells which were highly correlated (*Figure 3—figure supplement 2A and B*). We therefore took the overlap of peaks between biological replicates forward for downstream analyses, resulting in 13,542 peaks (*Figure 3—figure supplement 2C*; *Supplementary file 5*). These peaks are highly enriched in the KLF5 motif, demonstrating the validity of the dataset, and also in AP1(FRA1) and GATA (GATA6) motifs, which we have previously revealed in genome wide studies as implicated in OAC (*Britton et al., 2017*; *Rogerson et al., 2019*; *Figure 3—figure supplement 2D*). Focussing on the 97 genes that are upregulated in OAC and also downregulated after KLF5 depletion, 97% have a KLF5 ChIP-seq peak within 0.5 Mb of the TSS and the median distance between a KLF5 ChIP-seq peak and the TSS of all significantly downregulated genes was 11,975 bp (*Figure 3—figure supplement 2E*). In contrast, KLF5-binding regions are further away (>20 kb) from the TSS of genes that were either unaffected by KLF5 depletion or whose expression was increased. This is indicative of direct activation by KLF5. An example gene is *CDC25B* which harbours multiple KLF5 ChIP-seq peaks surrounding its locus (*Figure 3H*). Collectively, these results suggest a direct activator role of KLF5 in controlling cell cycle genes in OAC.

## The KLF5 cistrome is reconfigured during the progression from BO to OAC

Having determined a role for KLF5 in controlling cell-cycle-associated gene expression in OAC cells, we sought to determine the mechanism through which KLF5 acquires these functions. We first asked whether KLF5 expression changes in the transition from BO to OAC, however no increase in expression was found (*Figure 4A*). An alternative mechanism might be through redistributing the binding of KLF5 to different regulatory elements in OAC. We therefore hypothesised that KLF5 is active in both BO and OAC but may regulate specific genes in OAC by binding at different loci.

CP-A cells are derived from non-dysplastic BO and do not exhibit strong tumourigenic properties (*Lin et al., 2012*) so we compared KLF5 expression in BO-derived CP-A and OAC-derived OE19 cells and found that KLF5 is expressed at similar levels in (*Figure 4—figure supplement 1A*). We therefore we used these cell-lines to model KLF5 activity in BO and OAC. To gain a more comprehensive view of KLF5 function, we performed ChIP-seq for KLF5 in CP-A cells and used spike-in normalisation to better assess differential binding relative to OE19 cells. Anti-KLF5 antibodies precipitated KLF5 in CP-A cells (*Figure 4—figure supplement 1B*) and biological replicates were highly reproducible (*Figure 4—figure supplement 1C*). We took the overlap of peaks between biological replicates forward for downstream analyses, resulting in 13,526 peaks (*Figure 4—figure supplement 1D*; *Supplementary file 5*). Motif analysis showed high enrichment of the KLF5 motif further demonstrating the quality of the data (*Figure 4—figure supplement 1E*). KLF5 peaks from CP-A and OE19 cells were merged, generating a combined peak set of 21,353 peaks. Differential binding analysis revealed an altered KLF5 binding profile between CP-A and OE19 cells (*Figure 4B,C*

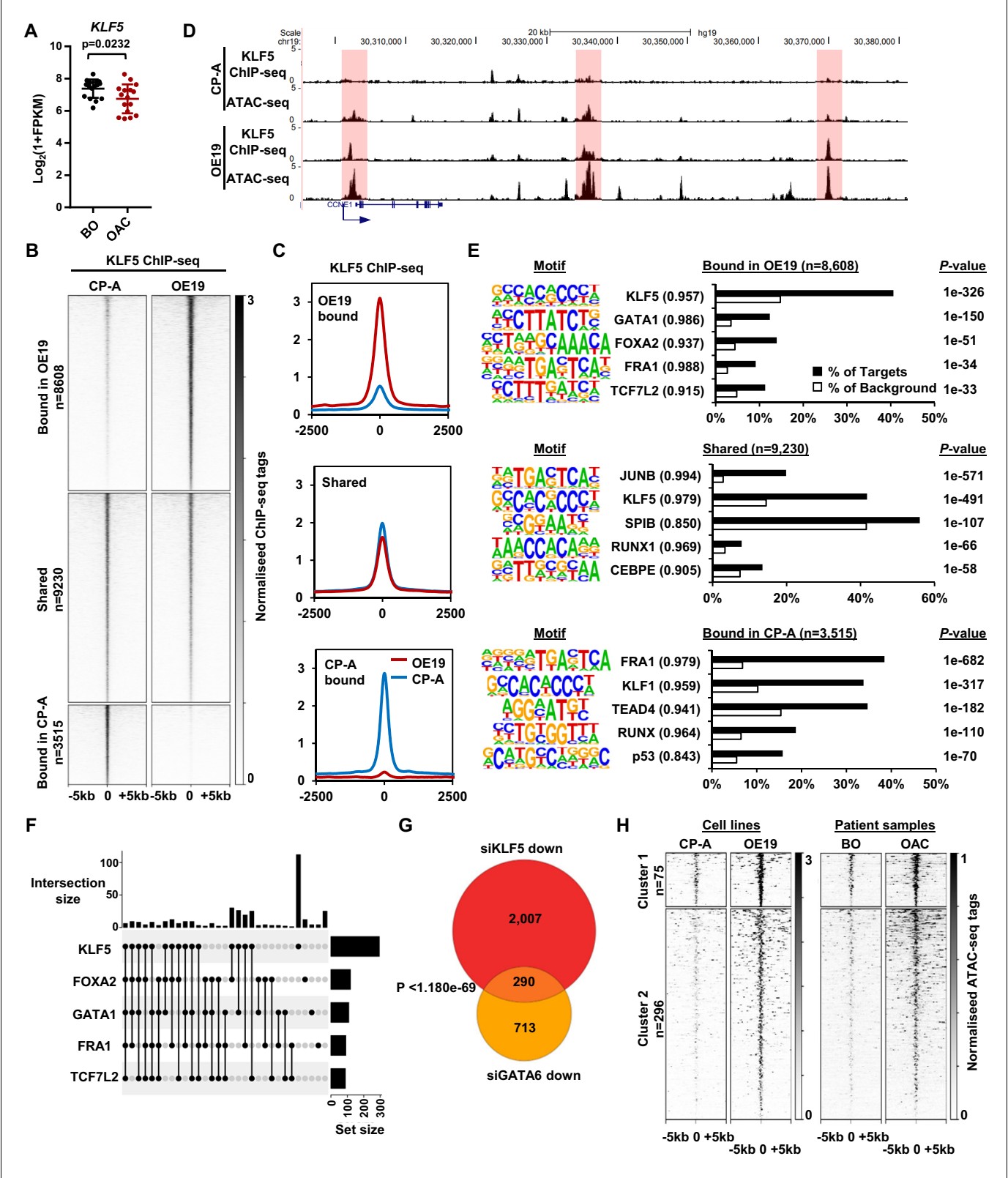

**Figure 4.** KLF5 binds to distinct regions in OE19 cells. (**A**) Expression $Log_2(1+FPKM)$ of *KLF5* in BO and OAC tissue. (**B**) Heatmap of KLF5 ChIP-seq signal at regions (peak centre ±5 kb) significantly bound in OE19 only (+2x; $Q$ < 0.05), shared regions (no significant change) and regions bound in CP-A only (−2x; $Q$-value <0.05). (**C**) Tag density plot of KLF5 ChIP-seq signal at regions (peak centre ±2.5 kb) bound in OE19 only, shared regions and regions bound in CP-A only. (**D**) Genome browser tracks showing KLF5 ChIP-seq and ATAC-seq in CP-A and OE19 cells at the *CCNE1* locus.
*Figure 4 continued on next page*

*Figure 4 continued*

Differential bound regions are highlighted in red. (E) Bar chart of percentage targets and percentage background of de novo discovered motifs at regions bound in OE19 only, shared regions and regions bound in CP-A only. De novo motifs, called transcription factor with match scores and *P*-values shown. (F) UPSET plot of DNA motifs found in 371 KLF5 binding regions that are specific to OE19-specific binding regions that are located within loci (+/- 250 kb) containing genes upregulated in OAC and downregulated with KLF5 depletion. The motifs identified in E (KLF5, GATA1, FOXA2, FRA1 and TCF7L2) found within each peak are shown. (G) Venn diagram showing the overlap in genes downregulated in OE19 cells following treatment with siRNAs targeting KLF5 and GATA6. (H) Heatmap of ATAC-seq signal at the KLF5 binding regions from (F) in the indicated cell lines (left) or patient derived tissue (right). Regions were subject to *k*-means hierarchical clustering (*k* = 2). See also *Figure 4—figure supplements 1* and *2*. The online version of this article includes the following figure supplement(s) for figure 4:

**Figure supplement 1.** KLF5 ChIP-seq analysis in CP-A cells.
**Figure supplement 2.** Integrative analysis of KLF5 ChIP-seq data in OE19 and CP-A cells.

*Supplementary file 5*): 8608 peaks show more binding in OE19 cells (40%), 9230 peaks are shared between the two cell-lines (44%) and 3515 show more binding in CP-A cells (16%). An example locus is *CCNE1*, which demonstrates increased KLF5 binding in OE19 cells at the promoter and putative enhancers associated with open chromatin regions (*Figure 4D*). Reciprocally, specific binding of KLF5 in CP-A cells is evident at the *NKX3-1* locus and common binding of KLF5 in both CP-A and OE19 cells is evident at the *JAG1* locus (*Figure 4—figure supplement 1F*).

We next asked whether the regions that exhibit differential binding are enriched for specific transcription factor motifs. Regions that are bound by KLF5 in OE19 cells are enriched in motifs for KLF, GATA, Forkhead, AP-1 and TCF transcription factors, whereas regions bound by KLF5 in CP-A cells are enriched for a different set of motifs with, TEAD, RUNX and p53 transcription factor families in addition to KLF and AP-1 motifs detected (*Figure 4E*; *Supplementary file 6*). These results are in-keeping with our previous work showing AP-1 and GATA6 functionality in OAC (*Britton et al., 2017*; *Rogerson et al., 2019*). Regions specifically bound by KLF5 in OAC cells also exhibited increased accessibility in OE19 cells and importantly, accessibility is also elevated around these binding sites in OAC tissue (*Figure 4—figure supplement 2A*). These findings are therefore consistent with a broad role of KLF5 in OAC.

To further probe the potential biological significance of the differentially bound KLF5 regions, we associated these with the nearest gene and determined the enriched GO terms for genes associated with cell-type-specific KLF5 peaks that also show preferential expression in BO or OAC. OE19-specific KLF5-binding events are associated with genes involved in 'cell division' control, whereas CP-A-specific KLF5 binding is associated with 'epithelial cell differentiation' (*Figure 4—figure supplement 2B*). The latter observation is consistent with the potential loss of cell identity in OAC. However, since oncogenic events during the progression from BO to OAC are poorly understood, we decided to focus on regions that acquire KLF5 binding in OE19 cells. To relate specific KLF5-binding events to gene expression changes, we took the set of 97 genes that are upregulated in OAC and downregulated by KLF5 depletion (i.e. activated by KLF5; *Figure 3E*) and found that there are 371 OE19-specific KLF5-binding peaks within a 0.5 Mb locus centred on the TSS. To further explore how KLF5 activates these genes in OAC, we assessed the transcription factor binding motif distribution (identified in *Figure 4E*) within this set of OE19-specific KLF5 peaks. We detected KLF5 binding motifs in 257/371 of these regions, and strikingly, 56% (145/257) of the peaks also house a mixture of FOXA, AP-1, GATA and TCF motifs, in addition to the KLF motif (*Figure 4F*; *Figure 4—figure supplement 2C*; *Supplementary file 7*). This suggests that KLF5 functions in a combinatorial manner with these other transcription factors to activate gene transcription during progression from BO to OAC. However, a large portion of these peaks (44%) contain only a KLF motif suggesting a more independent role for KLF5 in these regions (*Figure 4F*). To test whether these motif enrichments reflect transcription factor binding, we integrated our ChIP-seq data of transcription factors active in OAC (GATA6 and HNF4A; *Rogerson et al., 2019*), with KLF5-binding data. Since only GATA motifs are enriched in these regions we would expect co-binding with GATA6 and not HNF4A. We therefore compared ChIP-seq profiles for these transcription factors, and see extensive co-binding of GATA6 at these sites but no evidence of co-binding with HNF4A (*Figure 4—figure supplement 2D*). Finally, the predicted target gene co-regulation by KLF5 and GATA6 was validated by depletion of each factor in OE19 cells, which leads to a large significant overlap in downregulated genes (*Figure 4G*). However, this co-regulated gene set contains only two of the cell cycle associated genes regulated by KLF5,

suggesting that this combination of transcription factors is not directly involved in controlling this process. Turning back to the cell cycle genes directly activated in OAC through OAC-specific binding of KLF5, we tested whether KLF5 is relevant for their expression in CP-A cells. As expected from the lower KLF5-binding levels in these cells, depletion of KLF5 had little effect on these genes (*Figure 4—figure supplement 2E*), consistent with a newly acquired function in OAC cells.

To establish whether the 371 KLF5-bound regions that are associated with KLF sensitive genes are relevant to OAC, we turned back to our ATAC-seq data and clustered the data to reveal two clusters. One set of regions is already partially open in CP-A cells that increase in accessibility in OE19 cells (cluster 1) and another set are closed in CP-A cells and become more accessible (cluster 2) (*Figure 4H*, *Figure 4—figure supplement 2F*, left). Importantly, the same pattern of accessibility is evident using ATAC-seq signal from BO and OAC tissue (*Figure 4H*, *Figure 4—figure supplement 2F*, right). To identify any potential differences between these clusters, we performed motif analysis (*Figure 4—figure supplement 2G*; *Supplementary file 8*). The most common motif in both clusters were KLF motifs and the most striking difference is the large proportion of AP1 motifs specifically associated with cluster one suggesting a potential role for AP1 in priming binding of KLF5 to these regions.

Together, these results indicate an altered DNA-binding profile for KLF5 in BO and OAC, and this altered binding is associated with chromatin opening. This altered binding profile for KLF5 in OAC reflects a direct role in controlling genes involved in cell cycle.

## KLF5 converges with ERBB2 on cell cycle gene regulation and controls cell proliferation in OAC

Our results indicate a role of KLF5 in controlling increased cell cycle gene expression in OAC; however, it is unclear how this relates to genetic events that potentially impact on the same process. Genomic amplifications in signalling receptors are common in OAC, such as *ERBB2* (32% OAC have an *ERBB2* amplification; *Cancer Genome Atlas Research Network et al., 2017*) and occur during the transition from BO to OAC (*Stachler et al., 2015*). As the ERK pathway is implicated in promoting cell proliferation and is controlled by ERBB2, we investigated whether ERBB2 signalling impacts on KLF5-mediated gene regulatory events. First, we sought evidence for a link with transcription factor activity, and performed ATAC-seq on OE19 cells to investigate whether depletion of ERBB2 could alter chromatin accessibility. OE19 cells contain an amplification of the *ERBB2* locus (*Dahlberg et al., 2004*) and are dependent on ERBB2 for their proliferation (*Hong et al., 2012*). ERBB2 levels were efficiently reduced after 72 hr of siRNA treatment and phosphorylation of downstream targets (ERK and AKT) was reduced (*Figure 5—figure supplement 1A*). ATAC-seq data were reproducible and good quality (*Figure 5—figure supplement 1B and C*). We performed differential accessibility analysis, which identified 717 regions with decreased chromatin accessibility and 733 regions with increased accessibility (*Figure 5A*; *Supplementary file 9*). De novo motif analysis of the regions that exhibit reduced chromatin accessibility following ERBB2 depletion, revealed that the majority contain AP-1-binding motifs as expected from the established connections between ERK pathway signalling and AP1 transcription factors. However, the binding motif for KLF transcription factors was also detected, albeit in a subset of the regions (*Figure 5B*; *Supplementary file 10*). We then used our KLF5 ChIP-seq dataset from OE19 cells to validate KLF5 binding at regions with reduced chromatin accessibility following ERBB2 depletion (*Figure 5—figure supplement 1D*). These regions are relevant in the context of OAC as they also show increased chromatin accessibility in OAC tissue compared to BO (*Figure 5C*). The convergence of ERBB2 signalling on KLF5 transcription factor activity suggested that they might also converge on the same genes. We therefore carried out RNA-seq in OE19 cells treated with siRNA against ERBB2. The RNA-seq data were highly reproducible (*Figure 5—figure supplement 1E*) and resulted in 778 genes down- and 664 genes up-regulated (two-fold change; FDR < 0.05, FPKM > 1) (*Figure 5—figure supplement 1F*). There is a large, statistically significant overlap between directly activated KLF5 target genes and genes downregulated by *ERBB2* depletion. Moreover, a closer comparison reveals that the expression of the majority of the directly activated KLF5 target genes was reduced upon ERBB2 knockdown (*Figure 5D*). Most of these common target genes are cell cycle related. These results therefore indicate that ERBB2 and KLF5 converge on a similar set of regulatory regions to drive the expression of cell cycle regulatory genes. To establish whether ERBB2 can redistribute KLF5 binding and activate its target genes, we created BO-derived CP-A cell lines that stably over express ERBB2 to mimic the

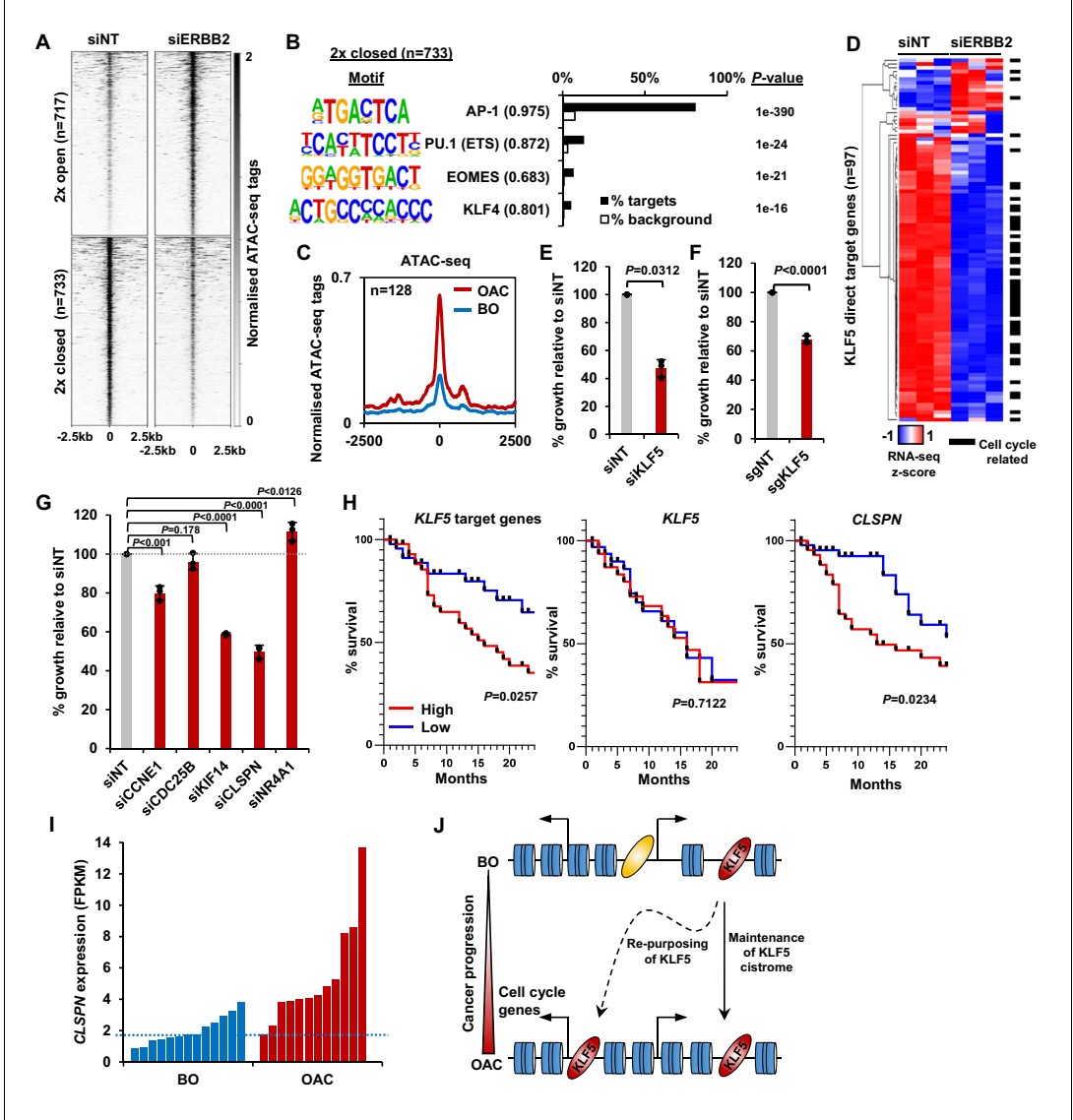

**Figure 5.** KLF5 controls cell proliferation in OAC. (**A**) Heatmap of ATAC-seq signal at regions (peak centre ±2.5 kb) with significantly differential accessibility (±2 x; Q < 0.05) in OE19 cells treated with siERBB2. (**B**) Bar chart of percentage targets and percentage background of de novo discovered motifs at regions closed with siERBB2 treatment in OE19 cells. De novo motifs, called transcription factor with motif match scores (brackets) and p-values are shown. (**C**) Tag density plot of ATAC-seq signal from BO and OAC tissue at regions that demonstrate KLF5 binding in OE19 cells and reduced chromatin accessibility in OE19 cells upon siERBB2 treatment. (**D**) Heatmap of z-score of expression of KLF5 direct target genes in OE19 cells treated with either siNT or siERBB2. Cell cycle related genes are indicated with a black bar. (**E**) Bar chart showing the % relative growth of OE19 cells treated with either siNT or siKLF5. p-Value is shown (n = 3). (**F**) Bar chart showing the % relative growth of OE19-dCas9-KRAB cells treated with either non-targeting guides or guides targeting the KLF5 TSS. p-Value is shown (n = 3). (**G**) Bar chart showing the % relative growth of OE19 cells treated with either siNT or siRNA against the indicated target genes. p-Values are shown (n = 3). (**H**) Kaplan-Meier curves of patient survival across 24 months for high (above median; red) or low (below median; blue) expression of the 97 KLF5 target genes (left), *KLF5* (middle) or *CLSPN* (right). p-Values are shown. (**I**) Expression levels (FPKM) of *CLSPN* expression in BO and OAC patient samples. (**J**) Model of KLF5 action in BO and OAC. KLF5 binds chromatin in BO and is re-purposed in OAC to bind and regulate cell-cycle-related genes. See also *Figure 5—figure supplements 1, 2* and *3*.

The online version of this article includes the following figure supplement(s) for figure 5:

**Figure supplement 1.** ERBB2 and KLF5 regulate an overlapping set of genes.
**Figure supplement 2.** ERBB2 overexpression drives growth factor independent proliferation and gene expression in BO-derived CP-A cells.
**Figure supplement 3.** KLF5 drives cell cycle progression in OE19 cells.

effect of amplification seen in OAC. These cells exhibit high levels of ERBB2 expression, maintain ERK and AKT activation in serum starved conditions (*Figure 5—figure supplement 2A*), and exhibit growth factor-independent proliferation (*Figure 5—figure supplement 2B*). Several cell cycle related genes that are activated by KLF5 in OAC cells are also activated by ERBB2 overexpression in BO cells (*Figure 5—figure supplement 2C*). However, we were unable to detect any increases in KLF5 occupancy at a panel of KLF5-binding regions associated with cell cycle genes (*Figure 5—figure supplement 2D*). These findings therefore reaffirm the convergence of ERBB2 signalling and KLF5 on the activation of a cell cycle gene signature but ERBB2 is not sufficient to trigger KLF5 redistribution.

Finally, we assessed whether defective KLF5-driven cell cycle gene regulation led to proliferative defects in OAC cells. We first depleted KLF5 in OE19 cells using siRNA which resulted in the reduction of KLF5 protein (*Figure 5—figure supplement 3A*), and the growth of cells was significantly impeded after siKLF5 treatment (*Figure 5E*). Second, we validated this growth defect by using CRISPR interference technology. Stable transfection of dCas9-KRAB and subsequent transfection of sgRNAs targeting the promoter of KLF5 (sgKLF5) into OE19 cells resulted in the reduction of KLF5 protein levels (*Figure 5—figure supplement 3B*). CRISPRi knockdown of KLF5 also significantly reduced the growth of OE19 cells (*Figure 5F*), mirroring the result with siKLF5. We further explored the role of KLF5 in cell growth and cell cycle progression by performing similar assays while perturbing KLF5 target genes (*CCNE1*, *CDC25B*, *KIF14*, *CLSPN* and *NR4A1*). All these genes showed significant reductions in expression upon siRNA treatment (*Figure 5—figure supplement 3C*). The growth of OE19 cells was significantly reduced with the treatment of siRNA against *CCNE1*, *KIF14* and *CLSPN* (*Figure 5G*). Knockdown of these genes also significantly altered cell cycle patterns, particularly knockdown of *CLSPN* which induced a prominent S-phase block (*Figure 5—figure supplement 3D*). These results provide more evidence for the role of KLF5 in the growth of cells and highlight the role of KLF5 target genes in this phenotype.

To assess whether the expression of KLF5 and its target genes has any clinical relevance, we sourced OAC expression and survival data (*Cancer Genome Atlas Research Network et al., 2017*) and plotted a survival of patients with high and low expression (±median) of KLF5 itself and KLF5 target genes up to 24 months (*Figure 5H*). Those with a higher expression of KLF5 showed no difference in patient survival, whereas patients with high target gene expression exhibited a significantly lower survival rate compared to those with low expression. This result is in keeping with the hypothesis that it is the activation of KLF5 target genes by its redistribution across chromatin, rather than its expression level that is important. It is noteworthy that CLSPN expression alone is predictive of increased patient survival (*Figure 5H*) and its enhanced expression in OAC compared to BO makes this a useful potential biomarker (*Figure 5I*).

Collectively, these results confirm the functional role of KLF5 in cell cycle control in OAC and convergence of action with the ERBB2 signalling pathway. This is clinically important as patients with highly expressed KLF5 target genes have a worse prognosis that those without.

## Discussion

Genome sequencing efforts of patients with BO and OAC have provided insights into the molecular causes of BO and OAC and show the mutational relationships between these disease states (*Ross-Innes et al., 2015*; *Stachler et al., 2015*). This has provided evidence for a model of OAC developing from BO. The molecular mechanisms involved in progression to OAC are poorly understood; however, BO offers a therapeutic window of opportunity to identify those more at risk of OAC development. In addition to genetic events, epigenetic changes and alterations to the chromatin landscape are also likely to play an important role in disease progression. Here, we demonstrate that there are marked changed in chromatin accessibility and associated gene expression, indicating active changes at the chromatin level during carcinogenesis. One of the major contributing factors to this change is the transcription factor KLF5. KLF5 is re-purposed in OAC cells and its chromatin-binding profile is massively rewired to drive increased expression of cell cycle associated genes (*Figure 5J*). Conversely, this rewiring results in the loss of KLF5 binding to many regulatory regions occupied in Barrett's cells. This loss is potentially associated with the loss of cell identity, and may also contribute to the development of the cancer phenotype.

Cell cycle deregulation is one of the key hallmarks of cancer (*Hanahan and Weinberg, 2011*) and here we uncovered a cell cycle gene expression signature, comprised of genes that are overexpressed in OAC. Recent research identified the cell-cycle as a perturbed pathway in OAC and suggested the possibility of CDK4/6 inhibitors as a therapeutic treatment (*Frankell et al., 2019*; *Mourikis et al., 2019*). We have previously uncovered a deregulated FOXM1 regulatory network active in OAC, a key regulator of late cell cycle gene expression (*Wiseman et al., 2015*). By integrating ATAC-seq data to identify upstream regulators of this signature, we also uncovered AP-1 and KLF5 as putative transcription factors in this process. We have previously identified AP-1 as an important factor in OAC (*Britton et al., 2017*) and others have shown an increase in AP-1 family transcription factors between non-dysplastic BO and low-grade dysplastic BO (*Maag et al., 2017*). What is less clear is the role of KLF5 in the progression of BO to OAC. KLF5 has been shown to have a tumour promoting function in pancreatic (*He et al., 2018*) and basal-like breast cancer (*Qin et al., 2015*). KLF5 is also frequently amplified in gastric cancer (*Chia et al., 2015*; *Zhang et al., 2018*) and has recently been shown to regulate gene expression in OAC in combination with other transcription factors, GATA6, ELF3 and EHF (*Chen et al., 2020*). This was reinforced by a recent study that identified KLF5 as a master transcription factor on which OAC cell-lines were dependent (*Reddy et al., 2019*). Paradoxically, KLF5 has been shown to have a tumour suppressor role in oesophageal squamous cell carcinoma (*Tarapore et al., 2013*) and breast cancer (*Chen et al., 2002*). The expression of the related protein, KLF4, together with three other genes, was able to stratify OAC from BO, albeit KLF4 expression is reduced in progression from BO to OAC (*Maag et al., 2017*).

Previous studies have begun to suggest a role for KLF5 in cell cycle control. For example, KLF5 binds to a *CCNE1* promoter proximal element in bladder cancer cells (*Pattison et al., 2016*) and KLF5 increases the expression of *Ccnb1* and *Mcm2* downstream of oncogenic Ras in fibroblasts (*Nandan et al., 2005*). Here, we provide evidence that KLF5 exhibits a widespread role; directly controlling cell proliferation through activation of cell cycle associated genes. We also show that reduction of KLF5 levels, or several of its target genes, in OAC cells impairs growth. Indeed, this is exemplified by *CLSPN* which may have therapeutic potential as its gene product, Claspin, has recently been shown to have a broader role in cancer cell viability by protecting cancer cells from replication stress (*Bianco et al., 2019*). KLF5 directly binds and regulates core cell cycle genes for example *CDC25B*, *CCNE1* and *MCM2*, some of which are cell cycle transcription factors for example E2F1, MYBL2, thus providing a mechanism for propagating its effects on cell cycle control. We also show KLF5 expression is almost unchanged between BO and OAC. By profiling KLF5 chromatin binding in BO and OAC cells, we have demonstrated an altered KLF5 binding profile. The regions bound by KLF5 specifically in OAC cells are enriched in motifs for several transcription factors, including the GATA family which suggests a combinatorial regulatory code. This is in keeping with our finding that there is extensive overlap between the binding of KLF5 and GATA6 which is reinforced by recent studies that show that KLF5 binds with GATA6 in OAC (*Chen et al., 2020*) and gastric cancer (*Chia et al., 2015*).

The overlap in regulatory potential with GATA6 provides a plausible link to one of the major genetic events that drive the BO to OAC transition. Our work also suggests a link to another major pathway that is activated through gene amplification in OAC, the ERBB2-driven RAS-ERK pathway. Knockdown of ERBB2 reduced the expression of many KLF5 target genes and KLF5 motifs were found at regions with reduced chromatin accessibility upon ERBB2 knockdown. However, ERBB2 overexpression in BO cells is insufficient to trigger KLF5 redistribution, indicating that other pathways contribute to KLF5 redistribution in OAC, but this needs further investigation. Nevertheless, it is clear that ERBB2 signalling and KLF5 activity converge on the same cell cycle genes and both are required for their activation, indicating functional synergy. The signalling pathways are more unclear in the context of BO, the precancerous precursor. We see enrichment of the TEAD motif only in CP-A cells and not OE19 cells, suggesting that KLF5 may be operating through the Hippo signalling pathway in BO. In other contexts, KLF5 has been shown to cooperate with TEAD transcription factors, downstream of YAP/TAZ (*Wang et al., 2015*) and KLF5 is stabilised by YAP in breast cancer cells (*Zhi et al., 2012*). Further work is needed to substantiate these links in BO.

In summary, we have used integrative analysis of RNA-seq and ATAC-seq from BO and OAC patient samples to uncover a cell cycle signature regulated by KLF5. Using a multi-omics approach, we found an oncogenic role of KLF5 in OAC, a transcription factor that has not been shown to be mutated, amplified and/or over-expressed in OAC. This study highlights the power of

supplementing expression data with genome-wide chromatin profiling methods such as ATAC-seq. This provides molecular insights into the mechanisms by which BO progresses to OAC and identifies a signature of transcription factor gene targets that have potential prognostic significance and could be used as biomarkers in the clinic.

## Materials and methods

### Cell lines, cell culture conditions and patient biopsies

OE19 and CP-A cells were purchased from ATCC and tested negative for mycoplasma. OE19 cells were maintained in Gibco RPMI 1640 (ThermoFisher, 52400) supplemented with 10% Gibco fetal bovine serum (ThermoFisher, 10270) and 1% Gibco penicillin/streptomycin (Thermo-Fisher,15140122). CP-A cells were cultured in keratinocyte serum free media (ThermoFisher, 17005042) supplemented with 5 µg/L EGF (ThermoFisher, 10450–013), 50 mg/L bovine pituitary extract (ThermoFisher, 13028014) and 10% Gibco fetal bovine serum (ThermoFisher, 10270) and 1% Gibco penicillin/streptomycin (ThermoFisher,15140122). Cell-lines were authenticated by STR profiling and routinely tested for mycoplasma.

OE19-dCas9-KRAB stable cells were generated by transfecting $1 \times 10^6$ OE19 cells with 7.5 µg Cas9 plasmid with guides targeting the *AAVS1* locus (Addgene #42230; 5'- GGGGCCACTAGGGA-CAGGAT-3') and 7.5 µg donor plasmid (pAAVS1-Puro-DNR; Origene GE100024) containing doxycy-cline inducible dCas9-KRAB with Fugene HD (Promega, E2311), as per manufacturer's instructions. After 7 hr, media was replaced and supplemented with 7.5 µM RS-1 (Sigma-Aldrich, R9782) and 1 µM SCR7 pyrazine (Sigma-Aldrich, SML1546), to promote homologous recombination and to inhibit non-homologous end joining respectively. Media was changed the next day and cells were selected with puromycin (0.75 µg/ml) for 14 days. Selected colonies were re-plated to grow single clones and clones screened for dCas9-KRAB protein expression by immunoblotting. OE19-dCas9-KRAB cells were cultured with 100 ng/mL doxycycline (Sigma-Aldrich, D3447) to induce dCas9-KRAB.

To create CP-A-ERBB2 (overexpressing ERBB2) and CP-A-empty (control) cells we first created the pHAGE-empty plasmid (pAS4940) by excising the ERBB2 coding sequence from pHAGE-ERBB2 (addgene #116734) using Xho1 (NEB, R0146S) followed by re-ligation of the vector. HEK293T cells were transfected with either pHAGE-ERBB2 or pHAGE-empty target plasmids, plus pMD2.G (Addg-ene, #12259), psPAX2 (Addgene, #12260) using Polyfect (Qiagen, 301107). Viral particles were pre-cipitated from media using PEG-it (System Biosciences, LV810A-1). CP-A-empty and CP-A-ERBB2 stable cells were generated by transfecting $1 \times 10^6$ CP-A cells with lentiviral particles containing either pHAGE-ERBB2 or pHAGE-empty using polybrene (EMD Millipore, TR-1003) at MOI of 1 for 24 hr. Transfected cells were grown for 2 days in full media before selection using puromycin (0.75 µg/ml) for 14 days.

Fresh frozen OAC 2 mm biopsies were obtained by consenting patients undergoing endoscopy. Tissue collection was granted by the ethics committee of Salford Royal NHS Foundation Trust (04/Q1410/57). Patient consent was obtained in written form and signed by the patient and doctor. Patient biological replicates are defined as separate patients, and cell-line biological replicates are defined as separate cell-lines cultures, processed at the same time.

### Protein extraction and immunoblotting

Cells were lysed directly in RIPA buffer and incubated on ice for 5 min. The lysate was then sonicated in a water bath sonicator (Diagenode Bioruptor) for 5 min, 30 s on/off and protein quantified using Pierce BCA Assay Kit (ThermoFisher, 23227). Lysates were supplemented with SDS-PAGE loading dye to a final concentration of 1x and boiled for 10 min. Equal amounts of protein were separated on a 10% polyacrylamide gel and transferred to a nitrocellulose membrane (GE life sciences, 1060002) using a Pierce Power Station (ThermoFisher). Membranes were blocked using Odyssey blocking buffer (Licor, 927–40000) and then incubated with antibodies against KLF5 (abcam, ab137676), Tubulin (Sigma-Aldrich, T9026), ERBB2 (ThermoFisher, MA5-14057), phospho-ERBB2 (Cell Signalling Technologies, 6942S), AKT (Cell Signalling Technologies, 2920S), phospho-AKT (Cell Signalling Technologies, 9106S), ERK1/2 (Cell Signalling Technologies, 4695S) or phospho-ERK1/2 (Cell Signalling Technologies, 9106S) overnight at 4°C. Membranes were incubated with IRDye sec-ondary antibodies (Licor, 925–32212, 925–32213) and imaged using a Li-Cor Odyssey scanner.

## RT-qPCR

RT-qPCR was carried out using QuantiTect SYBR Green RT-PCR Kit (Qiagen, 204243) using the primer pairs detailed in *Supplementary file 11*. Relative gene expression was calculated using the ΔΔCT method relative to levels of *GAPDH* mRNA.

## siRNA and sgRNA transfection

200,000 cells were plated on a six-well plate and incubated for 24 hr. 100 pmol either control non-targeting (siNT; Dharmacon, D-001810-10-0020), siKLF5 (Dharmacon, L-013571-00-0005), or siERBB2 (Dharmacon, L-003126-00-0005) SMARTpool siRNA was transfected per well using Lipofectamine RNAiMAX (Thermofisher, 13778150) as per the manufacturer's instructions and incubated for 72 hr. Modified full length sgRNAs were designed using E-CRISP (*Heigwer et al., 2014*; available at http://www.e-crisp.org/E-CRISP) using the *KLF5* TSS (±200 bp) as input and obtained from Synthego. 9 pmol of either control non-targeting sgRNA (5'- GUAAGGCUAUGAAGAGAUAC-3') or sgKLF5 pool (5'-GUGCGCUCGCGGUUCUCUCG-3'; 5'- AGGACGUUGGCGUUUACGUG-3'; 5'- GCG UCAAGUGUCAGUAGUCG-3') was transfected per well using Lipofectamine RNAiMAX (Thermofisher, 13778150). Media was changed after 72 hr for longer treatments.

## RNA extraction, RNA-seq processing and analysis

RNA was extracted from cells using a RNeasy RNA extraction Kit (Qiagen, 74136) and quality checked using Nanodrop 1000 (ThermoFisher). Paired-end RNA-seq libraries were generated using TruSeq stranded mRNA library kit (Illumina) and sequenced on a HiSeq 4000 platform (Illumina) by the University of Manchester Genomic Technologies Core Facility. Reads were trimmed using Trimmomatic v0.32 (*Bolger et al., 2014*) quality checked using FastQC (available at: http://www.bioinformatics.bbsrc.ac.uk/projects/fastqc) and aligned to RefSeq transcript annotation of GRCh37 (hg19) using STAR (*Dobin et al., 2013*). Reads aligned to chromosomes 1–22 and chromosome X were retained. The Cufflinks package v2.2.1 (*Trapnell et al., 2012*) was used to calculate gene expression levels using Cuffnorm, and to analyse differential gene expression using Cuffdiff. Default parameters were used in both instances. Significant gene expression changes were defined by a fold change of ±1.3 and a *Q*-value of <0.05. For ERBB2 knockdown experiments, counts for genes were determined using featureCounts (*Liao et al., 2014*). Log$_2$ transformed counts were obtained using DESeq2 variance stabilising transformation (VST) function.

## Crystal violet assay

200,000 cells were plated on a six-well plate and siRNA/sgRNA treatment started after 24 hr incubation. At specific time-points after treatment plates were washed with PBS and fixed with 4% paraformaldehyde for 10 min. Plates were then washed twice with PBS and kept at 4°C. Cells were then stained by first incubating plates at room temperature for 10 min in 0.1% Triton X-100 with gentle shaking and then incubated at room temperature for 30 min in 0.1% crystal violet (Sigma-Aldrich, HT90132) with gentle shaking. Plates were extensively washed with water multiple times and left to dry. The dye was solubilised with 10% acetic acid for 10 min with gentle shaking and absorbance was read at 590 nm. Values for siNT at each time-point were used as 100% growth.

## Propidium iodine staining assay

Cells were trypsinised and collected as a single-cell suspension, washed with cold PBS, then fixed in 70% ethanol and stored at −20°C for at least 2 hr. Cells were then resuspended in staining solution (50 µg/mL propidium iodide (Sigma, P4170), 100 µg/mL RNase (Sigma, R4642)) and incubated at room temperature for 30 min. Cells were analysed by the University of Manchester Flow Cytometry Core Facility on a LSRFortessa. Percentages of cells in each cell cycle phase were calculated using ModFit LT (http://www.vsh.com/products/mflt/).

## ATAC-seq, processing and analysis

Patient samples were processed as previously described (*Britton et al., 2017*) and omni-ATAC-seq was performed as previously described (*Corces et al., 2017*). ATAC-seq libraries (~8 per lane) were sequenced on a HiSeq 4000 platform (Illumina) by the University of Manchester Genomic Technologies Core Facility. Reads were quality checked using FastQC (available at: http://www.

bioinformatics.bbsrc.ac.uk/projects/fastqc) and aligned to GRCh37 (hg19) using Bowtie2 v2.3.0 (*Langmead and Salzberg, 2012*) with the following options: -X 2000 –dovetail. Unique reads (>q30) aligned to chromosomes 1–22 and chromosome X were retained. Peaks were called using MACS2 v2.1.1 (*Zhang et al., 2008*) with the following parameters: -q 0.01 –nomodel –`shift` −75 –`extsize` 150 -B –SPMR. Peaks called from individual samples were merged using mergePeaks.pl (using d = 250 parameter) from the HOMER package v4.9 (*Heinz et al., 2010*) and resized to peak summit ±250 bp to generate a peak set on which to perform differential accessibility analyses. Amplifications in patient biopsies were removed as described previously (*Denny et al., 2016*) using a fold change of 16. bedGraph files were converted into BigWig files using bedGraphtoBigWig and visualised in the UCSC Genome Browser (*Kent et al., 2002*).

For comparing BO and OAC ATAC-seq, the Cufflinks package v2.2.1 (*Trapnell et al., 2012*) was used to calculate chromatin accessibility levels using Cuffnorm, and differential chromatin accessibility was analysed using Cuffdiff. Default parameters were used in both instances. Significant chromatin accessibility changes were defined as a fold change of ±2 and a Q-value of <0.1.

For ERBB2 knockdown experiments differential accessibility was calculated using DESeq2 (*Love et al., 2014*). Alignment files of biological repeats were combined, peaks recalled and peaks from both conditions were then merged using using mergePeaks.pl (using d = 250 parameter) from the HOMER package v4.9 (*Heinz et al., 2010*) and resized to peak summit ±250 bp. featureCounts from the SUBread package (*Liao et al., 2014*) was used to count reads within peaks from ATAC-seq samples and these were used an input for DESeq2 to calculate differential binding using default settings. A linear fold change of ±2 and a Q-value of <0.05 were used as a cut-off for further analyses.

## ATAC-seq data visualisation

ATAC-seq fragment size was visualised using a custom python script. Correlation plots between technical replicates were visualised using multiBamSummary and plotCorrelation from the deepTools package (*Ramírez et al., 2016*). Tag density plots and heatmaps were also generated using computeMatrix and plotProfile or plotHeatmap tools from the deepTools package. ATAC-seq counts were also visualised using Morpheus (https://software.broadinstitute.org/morpheus/) and hierarchical clustering was performed with this software using 1-Pearson's correlation unless otherwise stated. Correlation plots of samples were visualised using the similarity matrix tool from Morpheus.

## De novo motif discovery

To analyse ATAC-seq or ChIP-seq peaks for enriched transcription factor motifs, genomic coordinates were analysed using findMotifsGenome.pl with –cpg –mask –size 200 -bg parameters from the Homer package (v4.7; *Heinz et al., 2010*). Background sequences were total accessible regions from all samples for ATAC-seq analysis and whole genome for ChIP-seq analysis.

## Gene set enrichment analysis and gene ontology analysis

Pre-ranked genes (ranked by $\log_2$(fold change)) were subject to gene set enrichment analysis from hallmark gene sets (h.all.v6.2) using GSEAPreranked from GSEA v3.0 (*Subramanian et al., 2005*). Gene ontology analysis was carried out using Metascape (*Zhou et al., 2019*; metascape.org).

## Footprinting analysis

To analyse footprinting signatures in ATAC-seq data the TOBIAS package was used (v0.5.1; *Bentsen et al., 2020*; available at https://github.molgen.mpg.de/loosolab/TOBIAS). Merged BAM files from each condition were processed using ATACorrect, footprint scores calculated using FootprintScores and differential footprinting analysis using BINDetect. Footprinting plots across identified footprints at TF motifs were plotted using plotProfile from the deepTools package (v2.5.0; *Ramírez et al., 2016*).

## ChIP-qPCR and ChIP-seq and analysis

ChIP-qPCR and ChIP-seq analysis was carried out as described previously (*Wiseman et al., 2015*). For ChIP-qPCR, $2.5 \times 10^6$ cells and 1 µg antibody were used, and analysed using a Rotor-Gene SYBR Green PCR Kit (Qiagen, 204074). For ChIP-seq, $1 \times 10^7$ cells, 5 µg target protein antibody, 1 µg Spike-in antibody (Active Motif, 61686) and 50 µl Protein A Dynabeads were used. 20 ng Spike-in

*Drosophila* chromatin (Active Motif, 53083) was supplemented to chromatin preps for Spike-in normalisation, as described previously (*Egan et al., 2016*). Normal rabbit IgG (Millipore, 12–370) antibody was used in parallel as a control. DNA libraries were prepared using TruSeq ChIP sample prep kit (Illumina) and sequenced on a HiSeq 4000 (Illumina) platform. Sequencing reads were aligned to GRCh37 (hg19) and dm6 using Bowtie2 v2.2.3 (*Langmead et al., 2009*). Reads aligning to the *Drosophila* genome were counted and used to generate scale factors. BAM files were then scaled to the sample with the lowest number of *Drosophila* reads. Only reads with a mapping quality >q30 were retained. Peak calling was performed on individual replicates using MACS2 v2.1.1 (*Zhang et al., 2008*) using default parameters with additional –SPMR parameter. bedGraph files were converted to bigwig using BedGraphtoBigWig script and visualised in the UCSC Genome Browser. The overlap of peaks between two biological replicates was calculated using BEDtools v2.26.0 (*Quinlan and Hall, 2010*) using bedtools intersect with default settings with -f 0.3 parameter. Peaks present in both datasets were taken forward for further analysis.

Differential binding analysis was performed using DESeq2 (*Love et al., 2014*). Overlaps from biological repeats were merged using bedtools merge to generate a final set of peaks. featureCounts from the SUBread package (*Liao et al., 2014*) was used to count reads within peaks from ChIP-seq samples and these were used an input for DESeq2 to calculate differential binding using default settings. A linear fold change of ±2 and a *Q*-value of <0.05 were used as a cut-off for further analyses.

## ChIP-seq visualisation

Heatmaps of ChIP-seq signal were generated using computeMatrix and plotHeatmap from the deepTools package (*Ramírez et al., 2016*). Tag density plots were generated using computeMatrix and plotProfile tools from the deepTools package. Correlation of biological replicates was visualised using multiBigwigSummary and plotCorrelation. Euler diagrams were generated using the Euler R package (available at eulerr.co).

## Principal component analysis

Principal component analysis was performed using the prcomp function in R (v3.5.1, R Core Team, 2018) using $\log_2$ transformed RNA-seq or ATAC-seq normalised counts. Principal component scores were then plotted in Excel.

## Patient survival analysis

Average expression of *KLF5* or KLF5 target genes (OAC upregulated and siKLF5 downregulated) was calculated per patient and patients were ranked by average target gene expression. The median was calculated and patients were classified as either above or under median expression. Survival (months) was plotted for each group and a Log-rank test was carried out using GraphPad Prism v8.

## Statistical analysis

To determine statistical significance between two groups, a Student's unpaired two-tail T- test was carried out using GraphPad Prism v7. To assess the changes in expression of a group of genes, a one-way ANOVA test was carried out in GraphPad Prism v7. To assess the significance of gene/region overlaps derived from sequencing data, a hypergeometric distribution test was carried out using the phyper function in R. p-values<0.05 were considered as significant.

## Datasets

All data were obtained from ArrayExpress, unless stated otherwise. ATAC-seq data from human BO, OAC tissue and OE19 cells were obtained from E-MTAB-5169 (*Britton et al., 2017*) and E-MTAB-6751 (*Rogerson et al., 2019*). BO and OAC RNA-seq data were obtained from E-MTAB-4054 (*Maag et al., 2017*) and European Genome-phenome Archive (EGA)(EGAD00001005915). GATA6 and HNF4A ChIP-seq were obtained from E-MTAB-6858 and siGATA6 RNA-seq from E-MTAB-6756. The Cancer Genome Atlas OAC ATAC-seq data were obtained from the GDC data portal (portal.gdc.cancer.gov; *Corces et al., 2018*).

## Data access

All sequencing data are deposited in ArrayExpress. Additional OAC ATAC-seq data are available at E-MTAB-8447 and additional BO and OAC RNA-seq data are available at E-MTAB-8584. siKLF5 RNA-seq data are available at E-MTAB-8446. KLF5 ChIP-seq data are available at E-MTAB-8568. siERBB2 ATAC-seq and RNA-seq data are available at E-MTAB-8576 and E-MTAB-8579 respectively. CP-A ATAC-seq data are available at E-MTAB-8994.

## Acknowledgements

We thank Guanhua Yan for excellent technical assistance and staff in the Bioinformatics, and Genomic Technologies core facilities and Mike Jackson in the Flow Cytometry facility. We also thank Elizabeth Ratcliffe for collection of BO samples. We also thank Nicoletta Bobola and Shen-Hsi Yang for critical appraisal of the manuscript. This work was funded by grants from the Wellcome Trust (103857/Z/14/Z) and Cancer Research UK (through the Manchester Cancer Research Centre).

## Additional information

### Funding

| Funder | Grant reference number | Author |
| --- | --- | --- |
| Cancer Research UK | Clinical PhD and PhD funding | Connor Rogerson<br>Edward Britton<br>Yeng Ang<br>Andrew D Sharrocks |
| Wellcome | Programme grant and studentship 103857/Z/14/Z | Samuel Ogden<br>Andrew D Sharrocks |

The funders had no role in study design, data collection and interpretation, or the decision to submit the work for publication.

### Author contributions

Connor Rogerson, Conceptualization, Formal analysis, Investigation, Methodology, Writing - original draft, Writing - review and editing; Samuel Ogden, Conceptualization, Formal analysis, Writing - review and editing; Edward Britton, Investigation, Writing - review and editing; The OCCAMS Consortium, Resources; Yeng Ang, Conceptualization, Resources, Supervision, Project administration, Writing - review and editing; Andrew D Sharrocks, Conceptualization, Resources, Supervision, Funding acquisition, Project administration, Writing - review and editing

### Author ORCIDs

Connor Rogerson (iD) http://orcid.org/0000-0002-1425-9668
Yeng Ang (iD) https://orcid.org/0000-0003-0496-6710
Andrew D Sharrocks (iD) https://orcid.org/0000-0002-6380-321X

### Ethics

Human subjects: Ethical approval was via the ethics committee of Salford Royal NHS Foundation Trust (04/Q1410/57). Patient consent was obtained in written form and signed by the patient and doctor.

### Decision letter and Author response

Decision letter https://doi.org/10.7554/eLife.57189.sa1
Author response https://doi.org/10.7554/eLife.57189.sa2

# Additional files

## Supplementary files

• Source code 1. ATAC fragment size visualisation.

• Supplementary file 1. Differentially expressed genes in OAC. Significantly (±1.5 x; Q-value <0.05) differentially expressed genes between BO (n = 13) and OAC (n = 12) (*Maag et al., 2017*).

• Supplementary file 2. Differentially accessible regions within ±250 kb of TSS of a DEG. (A) Total accessible regions from BO (n = 4) and OAC (n = 6) samples. (B) Significant differentially accessible open regions (+2x; Q-value <0.1). (C) Significant differentially accessible closed regions (−2x; Q-value <0.1).

• Supplementary file 3. DNA motifs enriched in OAC-specific open chromatin regions. Top ten motifs found by de novo motif discovery and their associated transcription factors that are enriched in 'open in OAC' (top) or 'closed in OAC' (bottom).

• Supplementary file 4. siKLF5 RNA-seq analysis. Significant differentially expressed genes with siKLF5 treatment (±1.3 x, Q-value <0.05)

• Supplementary file 5. KLF5 ChIP-seq datasets. (A) ChIP-seq peaks in OE19 cells. (B) ChIP-seq peaks in CP-A cells. (C) Differentially bound KLF5 ChIP-seq peaks (CP-A vs OE19).

• Supplementary file 6. De novo analysis of DNA motif enrichment in KLF5 ChIP-seq peak datasets.

• Supplementary file 7. (A) Frequency of KLF5, GATA1, FOXA2, FRA1 and TCF7L2 motifs within OE19 specific KLF5 ChIP-seq regions. one denotes present and 0 absent. (B) Overlaps of motifs and the basis of *Figure 4G* (A. KLF5; B. GATA1; C. FOXA2; D. FRA1; E. TCF7L2).

• Supplementary file 8. DNA motifs enriched in Cluster one and Cluster two regions. Top 10 motifs found by de novo motif discovery and their associated transcription factors that are enriched in cluster 1 (top) or cluster 2 (bottom).

• Supplementary file 9. Genomic coordinates of regions on OE19 cells that show a decrease in ATAC-seq signal upon treatment of siERBB2 for 72 hr.

• Supplementary file 10. De novo discovered motifs from regions that exhibit reduced chromatin accessibility upon treatment of siERBB2 for 72 hr. De novo motifs, % targets and % background, called transcription factor with match score and p-value are shown.

• Supplementary file 11. List of PCR primers used in RT-qPCR and ChIP-qPCR experiments.

• Transparent reporting form

## Data availability

All sequencing data are deposited in ArrayExpress. Additional OAC ATAC-seq data are available at E-MTAB-8447 and additional BO and OAC RNA-seq data are available at E-MTAB-8584. siKLF5 RNA-seq data are available at E-MTAB-8446. KLF5 ChIP-seq data are available at E-MTAB-8568. siERBB2 ATAC-seq and RNA-seq data are available at E-MTAB-8576 and E-MTAB-8579 respectively.

The following datasets were generated:

| Author(s) | Year | Dataset title | Dataset URL | Database and Identifier |
|---|---|---|---|---|
| Rogerson C, Ogden S, Britton E, The OCCAMS Consortium, Yeng A, Sharrocks AD | 2020 | Repurposing of KLF5 activates a cell cycle signature during the progression to Oesophageal Adenocarcinoma | https://www.ebi.ac.uk/arrayexpress/experiments/E-MTAB-8447 | ArrayExpress, E-MTAB-8447 |
| Rogerson C, Ogden S, Britton E, The OCCAMS Consortium, Yeng A, Sharrocks AD | 2020 | Repurposing of KLF5 activates a cell cycle signature during the progression to Oesophageal Adenocarcinoma | https://www.ebi.ac.uk/arrayexpress/experiments/E-MTAB-8446 | ArrayExpress, E-MTAB-8446 |
| Rogerson C, Ogden S, Britton E, | 2020 | Repurposing of KLF5 activates a cell cycle signature during the | https://www.ebi.ac.uk/arrayexpress/experiments/ | ArrayExpress, E-MTAB-8568 |

| Author(s) | Year | Dataset title | Dataset URL | Database and Identifier |
|---|---|---|---|---|
| The OCCAMS Consortium, Yeng A, Sharrocks AD | | progression to Oesophageal Adenocarcinoma | | E-MTAB-8568 |
| Rogerson C, Ogden S, Britton E, The OCCAMS Consortium, Yeng A, Sharrocks AD | 2020 | Repurposing of KLF5 activates a cell cycle signature during the progression to Oesophageal Adenocarcinoma | https://www.ebi.ac.uk/arrayexpress/experiments/E-MTAB-8584 | ArrayExpress, E-MTAB-8584 |
| Rogerson C, Ogden S, Britton E, The OCCAMS Consortium, Yeng A, Sharrocks AD | 2020 | Repurposing of KLF5 activates a cell cycle signature during the progression to Oesophageal Adenocarcinoma | https://www.ebi.ac.uk/arrayexpress/experiments/E-MTAB-8576 | ArrayExpress, E-MTAB-8576 |
| Rogerson C, Ogden S, Britton E, The OCCAMS Consortium, Yeng A, Sharrocks AD | 2020 | Repurposing of KLF5 activates a cell cycle signature during the progression to Oesophageal Adenocarcinoma | https://www.ebi.ac.uk/arrayexpress/experiments/E-MTAB-8579 | ArrayExpress, E-MTAB-8579 |
| Rogerson C, Ogden S, Britton E, The OCCAMS Consortium, Yeng A, Sharrocks AD | 2020 | Repurposing of KLF5 activates a cell cycle signature during the progression to Oesophageal Adenocarcinoma | https://ega-archive.org/datasets/EGAD00001005915 | EGA, EGAD00001005915 |
| Rogerson C, Ogden S, Britton E, The OCCAMS Consortium, Yeng A, Sharrocks AD | 2020 | Repurposing of KLF5 activates a cell cycle signature during the progression to Oesophageal Adenocarcinoma | https://www.ebi.ac.uk/arrayexpress/experiments/E-MTAB-8994 | ArrayExpress, E-MTAB-8994 |

The following previously published datasets were used:

| Author(s) | Year | Dataset title | Dataset URL | Database and Identifier |
|---|---|---|---|---|
| Britton E, Rogerson C, Mehta S, Li Y, Li X, The OCCAMS Consortium, Fitzgerald RC, Ang YS, Sharrocks AD | 2017 | Open chromatin profiling identifies AP1 as a transcriptional regulator in oesophageal adenocarcinoma | https://www.ebi.ac.uk/arrayexpress/experiments/E-MTAB-5169 | ArrayExpress, E-MTAB-5169 |
| Rogerson C, Britton E, Withey S, Hanley N, Ang Y, Sharrocks AD | 2019 | Identification of a primitive intestinal transcription factor network shared between esophageal adenocarcinoma and its pre-cancerous precursor state | https://www.ebi.ac.uk/arrayexpress/experiments/E-MTAB-6751 | ArrayExpress, E-MTAB-6751 |
| Rogerson C, Britton E, Withey S, Hanley N, Ang Y, Sharrocks AD | 2019 | Identification of a primitive intestinal transcription factor network shared between esophageal adenocarcinoma and its pre-cancerous precursor state | https://www.ebi.ac.uk/arrayexpress/experiments/E-MTAB-6756 | ArrayExpress, E-MTAB-6756 |
| Rogerson C, Britton E, Withey S, Hanley N, Ang Y, Sharrocks AD | 2019 | Identification of a primitive intestinal transcription factor network shared between esophageal adenocarcinoma and its pre-cancerous precursor state | https://www.ebi.ac.uk/arrayexpress/experiments/E-MTAB-6858 | ArrayExpress, E-MTAB-6758 |
| Maag JLV, Fisher OM, Levert-Mignon A, Kaczorowski DC, Thomas ML, Hussey DJ, Watson DI, Wettstein A, Bobryshev YV, Edwards M, Dinger ME, Lord RV | 2017 | Novel Aberrations Uncovered in Barrett's Esophagus and Esophageal Adenocarcinoma Using Whole Transcriptome Sequencing | https://www.ebi.ac.uk/arrayexpress/experiments/E-MTAB-4054 | ArrayExpress, E-MTAB-4054 |
| Corces MR, Granja JM, Shams S, Louie BH, Seoane JA, Zhou W, Silva TC, | 2018 | The chromatin accessibility landscape of primary human cancers | https://portal.gdc.cancer.gov/projects/TCGA-ESCA | GDC Data Portal, TCGA-ESCA |

Groeneveld C,
Wong CK, Cho SW,
Satpathy AT, Mum-
bach MR, Hoadley
KA, Robertson AG,
Sheffield NC, Felau
I, Castro MAA,
Berman BP, Staudt
LM, Zenklusen JC,
Laird PW, Curtis C,
The TCGA Network,
Greenleaf WJ,
Chang HY

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

# Appendix 1

**Appendix 1—key resources table**

| Reagent type (species) or resource | Designation | Source or reference | Identifiers | Additional information |
|---|---|---|---|---|
| Cell line (*H. sapiens*) | OE19 | ACACC | 96071721 | |
| Cell line (*H. sapiens*) | CP-A | ATCC | KR-42421 | |
| Cell line (*H. sapiens*) | OE19-dCas9-KRAB | This study | | OE19 transfected with vector to express dCas9-KRAB under doxycycline control |
| Cell line (*H. sapiens*) | CP-A-ERBB2 | This study | | CP-A stably overexpressing ERBB2 |
| Cell line (*H. sapiens*) | CP-A-empty | This study | | CP-A cells containing an empty vector control |
| Biological sample (*H. sapiens*) | Barrett's oesophagus biopsies | Salford NHS FT | | Freshly isolated from patients undergoing endoscopy |
| Biological sample (*H. sapiens*) | Oesophageal adenocarcinoma biopsies | Salford NHS FT | | Freshly isolated from patients undergoing endoscopy |
| Transfected construct (human) | SmartPool siRNA against KLF5 | Horizon discovery | L-013571-00-0005 | |
| Transfected construct (human) | SmartPool siRNA against ERBB2 | Horizon discovery | L-003126-00-0005 | |
| Transfected construct (human) | SmartPool non-targeting siRNA | Horizon discovery | D-001810-10-0020 | |
| Transfected construct (human) | Full length non-targeting guide RNA | Synthego | | 5'-GUAAGGCUAUGAAGAGAUAC-3' |
| Transfected construct (human) | Full length guide RNAs targeting KLF5 TSS | Synthego | | 5'-GUGCGCUCGCGGUUCUCUCG-3' <br> 5'-AGGACGUUGGCGUUUACGUG-3' <br> 5'-GCGUCAAGUGUCAGUAGUCG-3' |
| Antibody | Rabbit monoclonal KLF5 antibody | Abcam | ab137676 | (1:10000) for western blot; 5 ug for ChIP-seq |
| Antibody | Mouse monoclonal tubulin antibody | Sigma-Aldrich | T9026 | (1:2000) for western blot |
| Antibody | Spike-in antibody | Active Motif | 61686 | 1 ug for ChIP-seq |
| Antibody | Mouse monoclonal ErbB2 antibody | ThermoFisher | MA5-14057 | (1:1000) |
| Antibody | Mouse monoclonal AKT antibody | Cell signalling technology | 2920 | (1:2000) |
| Antibody | Rabbit monoclonal phosphor-Akt (S473) antibody | Cell signalling technology | 4060S | (1:2000) |

*Continued on next page*

*Appendix 1—key resources table continued*

| Reagent type (species) or resource | Designation | Source or reference | Identifiers | Additional information |
|---|---|---|---|---|
| Antibody | Rabbit monoclonal Erk1/2 antibody | Cell signalling technology | 4695S | (1:1000) |
| Antibody | Mouse monoclonal phosphor-Erk1/2 (T202, Y204) | Cell signalling technology | 9106S | (1:2000) |
| Antibody | Donkey anti-mouse secondary antibody (800CW) | Licor | 925 - 32212 | (1:10,000) |
| Antibody | Donkey anti-rabbit secondary antibody (700CW) | Licor | 925–32213 | (1:10,000) |
| Recombinant DNA reagent | pX330-U6-Chimeric_BB-CBh-hSpCas9 (plasmid) | Addgene | #42230 | AAVS guide RNA sequence 5'-GGGCCACTAGGGACAGGAT-3' |
| Recombinant DNA reagent | pAAVS1-Puro-TRE-dCas9-KRAB-DNR (plasmid) | This study | | pAS-4939 |
| Recombinant DNA reagent | pHAGE-ERBB2 | Addgene | 116734 | |
| Recombinant DNA reagent | pHAGE-empty | This study | | pAS-4940 |
| Recombinant DNA reagent | pMD2.G | Addgene | 12259 | |
| Recombinant DNA reagent | psPAX2 | Addgene | 12260 | |
| Sequenced-based reagent | Primers | This study | | See *Supplementary file 11* |
| Commercial assay or kit | Lipofectamine RNAiMAX | Thermofisher | 13778150 | |
| Commercial assay or kit | Fugene HD | Promega | E2311 | |
| Commercial assay or kit | QuantiTect SYBR Green RT-PCR Kit | Qiagen | 204243 | |
| Commercial assay or kit | RNeasy Plus Mini Kit | Qiagen | 74134 | |
| Commercial assay or kit | RNase-free DNase set | Qiagen | 79254 | |
| Commercial assay or kit | Ampure XP beads | Beckman Coulter Agencourt | A63881 | |
| Commercial assay or kit | TruSeq stranded RNA library kit v2 | Illumina | RS-122–2001 | |
| Commercial assay or kit | Nextera DNA library prep kit | Illumina | FC-121–1031 | |
| Commercial assay or kit | Nextera Index kit | Illumina | FC-121–1012 | |
| Commercial assay or kit | NEBNext high fidelity 2x PCR master mix | NEB | M0541 | |
| Commercial assay or kit | DNA Clean and Concentrator | Zymo | D4013 | |
| Commercial assay or kit | Polyfect | Qiagen | 301107 | |

*Continued on next page*

*Appendix 1—key resources table continued*

| Reagent type (species) or resource | Designation | Source or reference | Identifiers | Additional information |
|---|---|---|---|---|
| Commercial assay or kit | PEG-it | System Biosciences | LV810A-1 | |
| Commercial assay or kit | Polybrene | EMD Millipore | TR-1003 | |
| Chemical compound, drug | RS-1 | Sigma-Aldrich | R9782 | Used at final concentration 7.5 µM |
| Chemical compound, drug | SCR7 pyrazine | Sigma-Aldrich | SML1546 | Used at final concentration 1 µM |
| Chemical compound, drug | Doxycycline | Sigma-Aldrich | D3447 | Used at final concentration of 100 ng/mL |
| Chemical compound, drug | propidium iodide | Sigma | P4170 | Used at 50 µg/mL |
| Peptide, recombinant protein | RNase | Sigma | R4642 | Used at 100 µg/mL |
| Peptide, recombinant protein | EGF | ThermoFisher | 10450–013 | 5 µg/L |
| Peptide, recombinant protein | Bovine pituitary extract | ThermoFisher | 13028014 | Used at 50 mg/L |
| Software, algorithm | Trimmomatic | *Bolger et al., 2014* | V0.34 | http://www.usadellab.org/cms/?page=trimmomatic |
| Software, algorithm | Bowtie2 | *Langmead and Salzberg, 2012* | v2.3.0 | http://bowtie-bio.sourceforge.net/bowtie2/index.shtml |
| Software, algorithm | Star | *Dobin et al., 2013* | V2.5.4 | https://github.com/alexdobin/STAR |
| Software, algorithm | Macs2 | *Zhang et al., 2008* | v2.1.1 | https://github.com/taoliu/MACS |
| Software, algorithm | Cufflinks | *Tarapore et al., 2013* | v2.2.1 | http://cole-trapnell-lab.github.io/cufflinks/ |
| Software, algorithm | DEseq2 | *Love et al., 2014* | V1.22.2 | https://bioconductor.org/packages/release/bioc/html/DESeq2.html |
| Software, algorithm | TOBIAS | *Bentsen et al., 2020* | v0.5.1 | https://github.com/loosolab/TOBIAS |
| Software, algorithm | featureCounts | *Liao et al., 2014* | V1.6.2 | http://subread.sourceforge.net |
| Software, algorithm | FastQC | | v0.11.4 | https://www.bioinformatics.babraham.ac.uk/projects/fastqc/ |
| Software, algorithm | bedtools | *Quinlan and Hall, 2010* | v2.26.0 | https://bedtools.readthedocs.io/en/latest/ |
| Software, algorithm | DeepTools | *Ramírez et al., 2016* | V2.5.0 | https://deeptools.readthedocs.io/en/develop/ |

*Continued on next page*

*Appendix 1—key resources table continued*

| Reagent type (species) or resource | Designation | Source or reference | Identifiers | Additional information |
|---|---|---|---|---|
| Software, algorithm | GSEA | *Subramanian et al., 2005* | V3.0 | http://software.broadinstitute.org/cancer/software/gsea/wiki/index.php/Main_Page |
| Software, algorithm | Homer | *Heinz et al., 2010* | v4.9 | http://homer.ucsd.edu/homer/ |
| Software, algorithm | R | R Core Team (2018) | v3.5.1 | https://www.r-project.org/ |
| Software, algorithm | GraphPad Prism | | V8.0 | www.graphpad.com |
| Other | Crystal violet | Sigma Aldrich | HT90132 | Used at concentration of 0.1% |
| Other | Gibco RPMI 1640 | ThermoFisher | 52400 | |
| Other | Gibco fetal bovine serum | ThermoFisher | 10270 | |
| Other | Gibco penicillin/streptomycin | ThermoFisher | 15140122 | |
| Other | Keratinocyte SFM (1x) | ThermoFisher | 17005042 | |

