## [Decision Letter]

**Acceptance summary:**

In this study, the Sharrocks laboratory used an integrated genomics approach to reveal a key role of the transcription factor KLF5 in esophageal adenocarcinoma. The results show that KLF5 genome occupancy is rewired during the transformation process to drive the expression of key cell cycle genes required for proliferation.

**Decision letter after peer review:**

Thank you for submitting your article "Repurposing of KLF5 activates a cell cycle signature during the progression from a precursor state to oesophageal cancer" for consideration by *eLife*. Your article has been reviewed by three peer reviewers, including Irwin Davidson as the Reviewing Editor and Reviewer #1, and the evaluation has been overseen by Kevin Struhl as the Senior Editor.

The reviewers have discussed the reviews with one another and the Reviewing Editor has drafted this decision to help you prepare a revised submission.

As the editors have judged that your manuscript is of interest, but as described below that additional experiments are required before it is published, we would like to draw your attention to changes in our revision policy that we have made in response to COVID-19 (https://elifesciences.org/articles/57162). First, because many researchers have temporarily lost access to the labs, we will give authors as much time as they need to submit revised manuscripts. We are also offering, if you choose, to post the manuscript to bioRxiv (if it is not already there) along with this decision letter and a formal designation that the manuscript is 'in revision at *eLife*'. Please let us know if you would like to pursue this option. (If your work is more suitable for medRxiv, you will need to post the preprint yourself, as the mechanisms for us to do so are still in development.)

Summary:

This study uses integrated genomics approaches on both patient material and model cell lines to investigate changes in chromatin accessibility (ATAC-seq) and KLF5 genomic occupancy in Barrett's Esophagus (BO) compared to esophageal adenocarcinoma (OAC) in an attempt to explain the changes in gene expression that drive the transition from the pre-cancerous to cancerous state. The results show changes in ATAC-seq accessibility and redistribution of KLF5 genomic occupancy in particular around a collection of 97 genes that appear as key KLF5 targets involved in cell cycle and up-regulated in OAC. The study further shows that ERBB2 signaling, a gene frequently amplified during BO to OAC progression partly regulates the redistribution of KLF5 to sites regulating the cell cycle genes.

While the reviewers were convinced by the data showing the redistribution of KLF5 genomic occupancy between the BO and OAC states and appreciated the integration of patient data with model cell lines, one of the major conclusions is that this redistribution participates actively in the transition between the two states. This is implied by the loss of function data, but in the opinion of the reviewers, gain of function showing that ERBB2 overexpression drives KLF5 redistribution and the OAC gene expression program is also necessary. Other revisions are also required that involve data analyses and changes to the text. We will be happy receive a revised version of the manuscript once the revisions have been completed.

Essential revisions:

1) The authors show that siRNA mediated silencing of ERBB2 results in loss of KLF5 binding (not shown directly but through ATAC-seq) at specific sites selectively occupied in OAC. In a complementary experiment can the authors show that gain of ERBB2 expression in CP-A cells (or in another pertinent experimental system), enhances cell proliferation, promotes KLF5 redistribution and enhances expression of the key KLF5 cell cycle genes. Such gain of function experiments would strongly strengthen the conclusions from the loss of function experiments shown in the paper.

2) The authors should discuss a potentially opposing hypothesis for the role of KLF5 in OAC, that loss of KLF5 binding sites between CP-A and OE19 cells (Figure 4B) also has a direct role in gene expression changes and OAC phenotypes. Do these binding events correlate with genes with lower expression in OE19 cells (and OAC tissues relative to controls)? The focus on potential gain of function might occlude key gene regulatory processes resulting from a loss of KLF5 occupancy during the BO to OAC switch. Ultimately, this alternative hypothesis and detract from the overall impact of the findings.

3) The authors demonstrate that KLF5 and GATA6 knockdown affects a statistically enriched set of target genes in OE19 cells (Figure 4G), but this is only relevant if these same genes are NOT regulated by KLF5 and GATA6 in non-OAC cell lines (like the CP-A cells). If these genes are truly de novo targets in OAC, they should be KLF5/GATA6 independent in CP-A or other lines. Can the authors comment on this?

4) The authors should fully describe the RNA-seq results of the siERBB2 experiment. How many genes are de-regulated? If the genes in Figure 5D correspond to only a small subset of those de-regulated by ERBB2 silencing, then this would call their relevance into question and there is no indication of their statistical significance compared to the full repertoire of ERBB2 regulated genes.

---

## [Author Response]

[…] While the reviewers were convinced by the data showing the redistribution of KLF5 genomic occupancy between the BO and OAC states and appreciated the integration of patient data with model cell lines, one of the major conclusions is that this redistribution participates actively in the transition between the two states. This is implied by the loss of function data, but in the opinion of the reviewers, gain of function showing that ERBB2 overexpression drives KLF5 redistribution and the OAC gene expression program is also necessary. Other revisions are also required that involve data analyses and changes to the text. We will be happy receive a revised version of the manuscript once the revisions have been completed.

In the original manuscript, we demonstrated that ERBB2 and KLF5 converge on the same signature of cell cycle genes that are more active in OAC. In our conclusions, we were careful to state that this was convergence rather than direct pathway intersection. We used this as further evidence to demonstrate the importance of the KLF5 regulon for gene regulation OAC. However, the reviewers make a valid point that we should test whether the functional interaction if direct or indirect. We now demonstrate that overexpression of ERBB2 in a Barrett’s cell line (CP-A) is sufficient to drive growth factor-independent activation of a panel of cell cycle genes from the KLF5 regulon (new Figure 5—figure supplement 2C), consistent with its ability to drive growth factor-independent cell proliferation (Figure 5—figure supplement 2B). However, ERBB2 did not promote enhanced KLF5 binding to the regulatory regions of cell cycle genes in these cells (new Figure 5—figure supplement 2D). These results are consistent with ERBB2 and KLF5 converging on the same genes but with no direct link between ERBB2 overexpression and KLF5 redistribution. Importantly, we now also show that KLF5 does not regulate a panel of cell cycle genes in Barrett’s (CP-A) cells (new Figure 4—figure supplement 2E) which we previously showed to be regulated and bound by KLF5 in OAC cells. This provides further evidence for KLF5 specifically functioning in the context of OAC to drive cell cycle gene activation.

Thus, our original conclusions are still valid and we have now altered the Discussion to be more definitive about the lack of direct links between ERBB2 and KLF5 redistribution, the trigger for which still needs to be determined though future studies.

Essential revisions:1) The authors show that siRNA mediated silencing of ERBB2 results in loss of KLF5 binding (not shown directly but through ATAC-seq) at specific sites selectively occupied in OAC. In a complementary experiment can the authors show that gain of ERBB2 expression in CP-A cells (or in another pertinent experimental system), enhances cell proliferation, promotes KLF5 redistribution and enhances expression of the key KLF5 cell cycle genes. Such gain of function experiments would strongly strengthen the conclusions from the loss of function experiments shown in the paper.

See the discussion above. We have demonstrated that ERBB2 is sufficient to drive growth factor-independent cell proliferation of Barrett’s (CP-A) cells (new Figure 5—figure supplement 2A and B) and enhances expression of a panel of KLF5 target genes in these cells (see new Figure 5—figure supplement 2C). This is fully consistent with our loss of function experiments performed in OAC cells. However, we were unable to detect increases to KLF5 occupancy at a panel of regulatory regions (new Figure 5—figure supplement 2D) indicating that ERBB2 alone is insufficient to trigger KLF5 redistribution. In the original paper we concluded that KLF5 and ERBB2 converge on the same set of genes but not necessarily through direct interaction. This new data has allowed us to reaffirm this conclusion and rule out a causative effect of ERBB2 on KLF5 redistribution. This is not surprising given the complexities of cancer cell rewiring and future studies will be needed to investigate the underlying mechanism further. Note that our ATAC-seq data do point to a loss of KLF5 binding a subset of regions following ERBB2 inhibition but these are not relevant to the cell cycle phenotype we observe here.

2) The authors should discuss a potentially opposing hypothesis for the role of KLF5 in OAC, that loss of KLF5 binding sites between CP-A and OE19 cells (Figure 4B) also has a direct role in gene expression changes and OAC phenotypes. Do these binding events correlate with genes with lower expression in OE19 cells (and OAC tissues relative to controls)? The focus on potential gain of function might occlude key gene regulatory processes resulting from a loss of KLF5 occupancy during the BO to OAC switch. Ultimately, this alternative hypothesis and detract from the overall impact of the findings.

To address this issue, we examined the types of genes associated with the KLF5 regions that are lost in OAC cells and the resulting GO terms suggest a role in epithelial development which may have been retained in CP-A cells from an earlier stage (new Figure 4—figure supplement 2B). We added discussion of this to the paper and point out a potential role for this change (i.e. KLF5 binding loss) in the transition to OAC potentially by potentially losing developmental identity. Further work will be needed to further develop this hypothesis which is outside the scope of the current study.

3) The authors demonstrate that KLF5 and GATA6 knockdown affects a statistically enriched set of target genes in OE19 cells (Figure 4G), but this is only relevant if these same genes are NOT regulated by KLF5 and GATA6 in non-OAC cell lines (like the CP-A cells). If these genes are truly de novo targets in OAC, they should be KLF5/GATA6 independent in CP-A or other lines. Can the authors comment on this?

To address this point, we depleted KLF5 in CP-A cells and tested a panel of genes that are dependent on KLF5 for their activity in OE19 cells. No significant decreases in target gene expression were observed (new Figure 4—figure supplement 2E), demonstrating that they are KLF5-independent in Barrett’s cells but become dependent on KLF5 activity in OAC-derived OE19 cells. We have not tested GATA-dependency as this is not central to this paper on cell cycle functions of re-purposed KLF5, and the contribution of GATA-6 to the broader gene regulatory network warrants further extensive work that is beyond the scope of the current paper.

4) The authors should fully describe the RNA-seq results of the siERBB2 experiment. How many genes are de-regulated? If the genes in Figure 5D correspond to only a small subset of those de-regulated by ERBB2 silencing, then this would call their relevance into question and there is no indication of their statistical significance compared to the full repertoire of ERBB2 regulated genes.

We tested the significance of the overall overlap of KLF5 activated direct target genes (i.e. down with siKLF5) and ERBB2 activated genes (i.e. down with siERBB2 treatment) and show that this is highly significant whereas genes reciprocally regulated show a negligible, non-significant overlap (new Figure 5—figure supplement 1F). This reaffirms our conclusions that ERBB2 signalling and KLF5 converge on the same set of genes which are largely involved in cell cycle control (note that in Figure 5D, we do not take a fold cut off for the effect of ERBB2 to better illustrate the directionality of change).